# The orphan nuclear receptor Nurr1 is responsive to non-steroidal anti-inflammatory drugs

Sabine Willems[1], Whitney Kilu[1], Xiaomin Ni[1,2], Apirat Chaikuad[1,2], Stefan Knapp[1,2], Jan Heering[3] & Daniel Merk[1✉]

Nuclear receptor related 1 (Nurr1) is an orphan ligand-activated transcription factor and considered as neuroprotective transcriptional regulator with great potential as therapeutic target for neurodegenerative diseases. However, the collection of available Nurr1 modulators and mechanistic understanding of Nurr1 are limited. Here, we report the discovery of several structurally diverse non-steroidal anti-inflammatory drugs as inverse Nurr1 agonists demonstrating that Nurr1 activity can be regulated bidirectionally. As chemical tools, these ligands enable unraveling the co-regulatory network of Nurr1 and the mode of action distinguishing agonists from inverse agonists. In addition to its ability to dimerize, we observe an ability of Nurr1 to recruit several canonical nuclear receptor co-regulators in a ligand-dependent fashion. Distinct dimerization states and co-regulator interaction patterns arise as discriminating factors of Nurr1 agonists and inverse agonists. Our results contribute a valuable collection of Nurr1 modulators and relevant mechanistic insights for future Nurr1 target validation and drug discovery.

[1] Institute of Pharmaceutical Chemistry, Goethe University Frankfurt, Max-von-Laue-Str. 9, 60438 Frankfurt, Germany. [2] Structural Genomics Consortium, BMLS, Goethe-University Frankfurt, Max-von-Lause-Str. 15, 60438 Frankfurt, Germany. [3] Fraunhofer Institute for Molecular Biology and Applied Ecology IME, Branch for Translational Medicine and Pharmacology TMP, Theodor-Stern-Kai 7, 60596 Frankfurt, Germany. ✉email: merk@pharmchem.uni-frankfurt.de

Nuclear receptor related 1 (Nurr1, NR4A2), a member of the nerve growth factor-induced β subfamily of orphan nuclear receptors[1,2], is a neuroprotective transcription factor mainly found in dopaminergic neurons. Levels of Nurr1 are diminished in Parkinson's disease (PD) patients and midbrain dopaminergic neuron development is Nurr1 dependent. Moreover, Nurr1 knock-out in mice in mature dopamine neurons resembled the progressive pathology seen in early stage of PD[3] suggesting therefore Nurr1 as promising target in PD treatment.

Nurr1 was originally considered as a ligand-independent nuclear receptor (NR) due to its closed ligand-free conformation and its high constitutive activity[1]. However, recent reports indicate that Nurr1 activity can be modulated with small-molecule ligands. Despite the lack of a canonical binding site in the apo structure of the Nurr1 ligand-binding domain (LBD), dynamic NMR, hydrogen deuterium exchange, and mutagenesis studies have defined two potentially overlapping ligand-binding regions within the Nurr1 LBD for unsaturated fatty acids (UFAs) and amodiaquine type ligands[4–6]. These putative, highly solvent accessible binding sites[5] are located on the LBD surface around helix 3, which is distant from helix 12 that often has a canonical activation function in other nuclear receptors. In addition, X-ray structures of the Nurr1 LBD in complexes with prostaglandin A1 (PGA1, PDB-ID: 5Y41[7]), prostaglandin A2 (PGA2, PDB-ID: 5YD6[8]), and 5,6-dihydroindole (DHI, PDB-ID: 6DDA[9]) have been reported recently, in which the ligands are covalently bound in an induced pocket between helices 5 and 12 that is not present in the apo structure.

Some recent studies have reported a number of Nurr1 ligands, demonstrating the potential of modulation of the receptor activity by small molecules. UFAs such as docosahexaenoic acid (DHA), arachidonic acid, linoleic acid, and oleic acid were identified as the first natural Nurr1 ligands[4,5,10]. Neutral antagonistic effects of DHA and other UFA metabolites without intrinsic activity have been suggested[4,5,10], however, their cellular effects on Nurr1 activity and potential biological relevance remain elusive. Conversely, the prostaglandins E1 and A1 have recently been reported as naturally occurring Nurr1 activators and found to exhibit Nurr1-dependent neuroprotective effects[7]. A series of isoxazolopyridinones has been described as synthetic Nurr1 activators, albeit with weak activation efficacies not exceeding 2-fold activation[11]. Recently, the antimalarials amodiaquine (AQ) and chloroquine (CQ) with chloroquinoline scaffold were also discovered as Nurr1 modulators with a slightly higher efficacy (~3-fold activation) but markedly lower potency[6]. While the putative neutral Nurr1 antagonism of UFAs requires further characterization, the prostaglandins E1 and A1 as well as all synthetic Nurr1 ligands activate the nuclear receptor and further promote its already high basal transcriptional inducer activity. The limited efficacies of these Nurr1 activators coupled with the lack of inverse Nurr1 agonists that suppress the receptor's intrinsic activity prompt further efforts in the search for potent Nurr1 ligands that can be used as a tool for biological studies of the receptor's roles in health and disease.

Here we report the discovery of several structurally diverse cyclooxygenase (COX) inhibitors as Nurr1 modulators with distinct activity profiles. The tricyclic compounds oxaprozin, valdecoxib, and parecoxib, as well as meloxicam markedly diminish constitutive Nurr1 activity and thereby act as inverse Nurr1 agonists. Meclofenamic acid (MFA) is characterized as selective Nurr1 modulator with agonist and inverse agonist properties. Together with the previously reported Nurr1 agonists AQ and CQ, these Nurr1 ligands serve as chemical tools to study the receptor's mode of action. Both classes of binders with converse effects demonstrate that Nurr1 activity can be modulated by small-molecule ligands in a bidirectional fashion. The use of

Nurr1 agonists and inverse agonists in cofactor recruitment assays reveals potential interactions between Nurr1 and several nuclear receptor co-regulators, including NCoR-1, NCoR-2, NRIP1, and NCoA6, in a ligand-concentration-dependent manner. Moreover, the Nurr1 modulators interestingly affect heterodimerization between Nurr1 and RXRα as well as Nurr1 homodimerization with distinctive profiles. Cross-titration additionally demonstrates that Nurr1 can be modulated simultaneously by the different types of binders suggesting distinct binding sites. These results contribute to our understanding of Nurr1 biology, and may open new avenues for pharmacological Nurr1 modulation.

## Results

**Nurr1 reveals binding site close to the activation function.** The recently published co-crystal structures of the Nurr1 LBD bound to prostaglandin A1 (PGA1; PDB-ID: 5Y41[7]) and the dopamine metabolite DHI (PDB-ID: 6DDA[9]) reveal a ligand-binding site differing from typical nuclear receptor LBDs (Fig. 1). While most nuclear receptors accommodate ligands inside the three-layer sandwich LBD structure between helices 2, 3, 5, 6, 7, 11, and 12, this pocket is located between helices 5 and 12 in Nurr1 and it is closer to the LBD surface. In comparison to the Nurr1 apo structure (PDB-ID: 1OVL[1]), the binding of PGA1 and DHI requires an outward movement of helix 12 by ~10–21° from its closed position, essential for activation, to create the ligand-binding pocket. Such movement upon ligand binding affects and may weaken the salt bridge between Lys590 (H12) and Glu440 (H5) observed in the ligand-free state, suggesting that this region in proximity to helix 5 and 12 might provide a handle to modulate transcriptional inducer activity of Nurr1. In parallel, a different binding site has been postulated for the Nurr1 activator AQ, and is located at the Nurr1 LBD surface between helices 3 and 6[6]. Mutagenesis experiments support this assumption indicating that Nurr1 has two, potentially independent, ligand-binding pockets that enable modulation by small-molecule ligands.

**Cyclooxygenase inhibitors modulate NR4A receptors in vitro.** Inspired by the processed COX metabolite PGA1 as Nurr1 ligand[7], we hypothesized that COX inhibitors might bind to Nurr1, prompting us to screen a comprehensive collection of drug-approved COX-1 and COX-2 inhibitors for Nurr1 modulation. To capture also selectivity or promiscuity among the closely related NR4A receptors nerve growth factor-induced β (Nur77, NR4A1) and neuron-derived orphan receptor 1 (NOR1, NR4A3), we included all three NR4A receptors in this primary screen. For this, we employed cellular hybrid reporter gene assays based on chimeric receptors composed of the human NR4A receptor LBD and the DNA binding domain of Gal4 from yeast. A Gal4-sensitive firefly luciferase reporter construct served as reporter gene and constitutively expressed renilla luciferase (SV40 promoter) was employed to normalize for transfection efficiency and to monitor test compound toxicity. In agreement with their natural behavior[1], the chimeric Gal4-NR4A receptors revealed strong ligand-independent, intrinsic transcriptional inducer activity in this setting. AQ and CQ in conformity with literature[6] activated Gal4-Nurr1 with $EC_{50}$ values in an intermediate micromolar range and up to 3.6-fold activation efficacy. To exclude non-specific effects in this cellular test system, control experiments were performed for all compounds affecting NR4A receptor activity, in which the potent ligand-independent transcriptional inducer Gal4-VP16[12] replaced Gal4-NR in the assay setup. Only compounds affecting NR4A-dependent but not

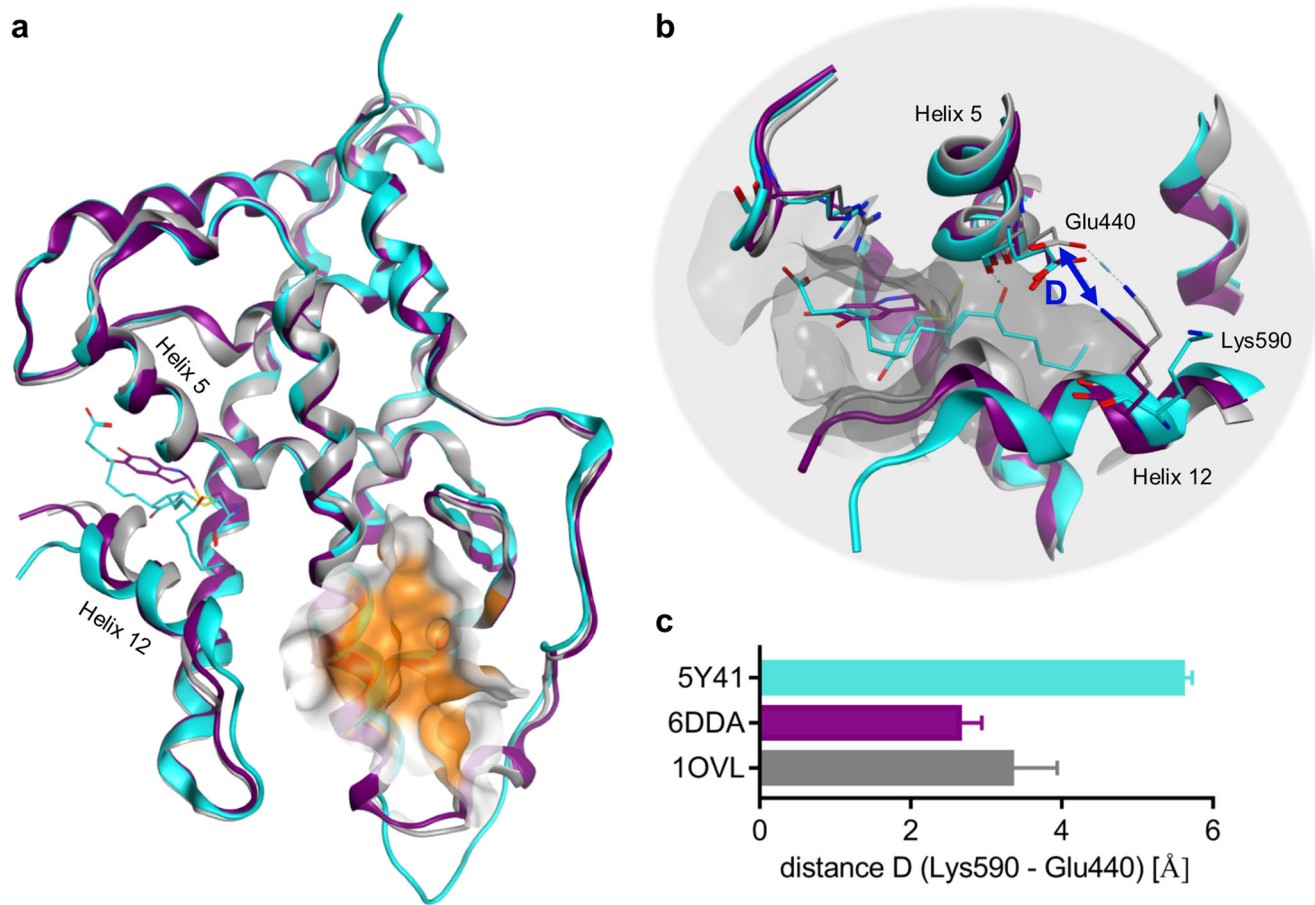

**Fig. 1 Structural differences in apo and ligand-bound Nurr1 LBD. a** Superposition of the Nurr1 LBD in apo state (PDB: 1OVL[1], gray), Nurr1 LBD bound to prostaglandin A1 (PGA1; PDB: 5Y41[7], cyan) and to dopamine metabolite 5,6-dihydroindole (DHI; PDB: 6DDA[9], purple). The proposed binding region for amodiaquine type ligands[6] is highlighted in orange. **b** In apo state, helix 12 forms several contacts with helix 5, including a prominent salt-bridge interaction between Lys590 and Glu440 likely stabilizing a transcriptionally active conformation of the activation function. Ligand binding causes an outward swing of helix 12 for creating the binding pocket, which interferes helix 5 and 12 interaction as demonstrated for example by different Lys590-Glu440 distances in different structures (shown in **c**). **c** Distances (D) between Lys590 ($NH_3^+$) and Glu440 ($CO_2^-$) measured in the different Nurr1 LBD X-ray structures. Compared with the apo state (1OVL, gray), the DHI bound structure (6DDA, purple) revealed a diminished distance, while the distance was increased in the PGA1 bound structure (5Y41, cyan). Bars represent the mean ± SD distances measured in the subunits of the respective structures.

VP16-dependent reporter activity were further considered as NR4A modulators.

Thirty-nine non-steroidal anti-inflammatory drugs (NSAIDs) covering all chemotypes of drug-approved COX inhibitors were initially screened in the Gal4-NR4A reporter gene assays at concentrations of 10 and 30 μM for NR4A modulatory activity (Fig. 2a). Meclofenamic acid, clonixin, and tiaprofenic acid enhanced Nurr1-dependent reporter activity (>1.5-fold activation) indicating Nurr1 agonism. Control experiments on Gal4-VP16 revealed non-specific activity of clonixin and tiaprofenic acid whereas the activity of meclofenamic acid was clearly Nurr1 mediated (Fig. 3e). Full dose–response characterization of meclofenamic acid resulted in an $EC_{50}$ value of 4.7 ± 0.1 μM and 3.52 ± 0.05-fold maximum activation on Gal4-Nurr1 (Fig. 3a, Table 1). For aceclofenac, the primary screen indicated Nurr1 repressor activity, however, full characterization conversely revealed this compound as a potent Nurr1 agonist ($EC_{50}$ = 2.5 ± 0.1 μM, 1.99 ± 0.07-fold max. activation), suggesting that the repressive effect at high concentration is likely due to inhibition of firefly luciferase. Nurr1 repression (<0.80-fold activation) was observed for oxaprozin, valdecoxib, and parecoxib at 10 or 30 μM. Control experiments on Gal4-VP16 indicated no non-specific reporter suppression (Fig. 3e, Supplementary Fig. 1b) and

dose–response characterization confirmed inverse agonism for all three compounds (Fig. 3a, Supplementary Fig. 1c). Oxaprozin was the strongest Nurr1 repressor and diminished Nurr1 activity to 0.26 ± 0.08-fold minimum activation with an $IC_{50}$ value of 40 ± 6 μM. Meloxicam demonstrated also strong repressor efficacy with a higher $IC_{50}$ value while valdecoxib and parecoxib were less effective but had slightly lower $IC_{50}$ values on Nurr1.

Neither the Nurr1 modulators discovered in our screening nor the previously known ligands AQ and CQ were selective for Nurr1 over the related NR4A receptors. AQ (100 μM) revealed even stronger activation efficacy on Nur77 and NOR1 while CQ (100 μM) and MFA (10 μM) as well as inverse agonists parecoxib (30 μM) and oxaprozin (50 μM) caused similar modulation of all three NR4A receptors (Fig. 2c, Supplementary Table 1). Generally, the activity of NSAIDs was similar on all three NR4As with few exceptions (Fig. 2a, Supplementary Table 1). Lornoxicam and mofezolac demonstrated inverse agonism on Nur77 and NOR1 (Supplementary Fig. 2) without affecting Nurr1 activity. Considering also the structural similarity of mofezolac and oxaprozin, this suggests that selectivity amongst NR4As is achievable despite their close similarity.

To profile also the selectivity of Nurr1 modulators MFA, parecoxib and oxaprozin (Fig. 2d; aceclofenac, meloxicam, and

 3

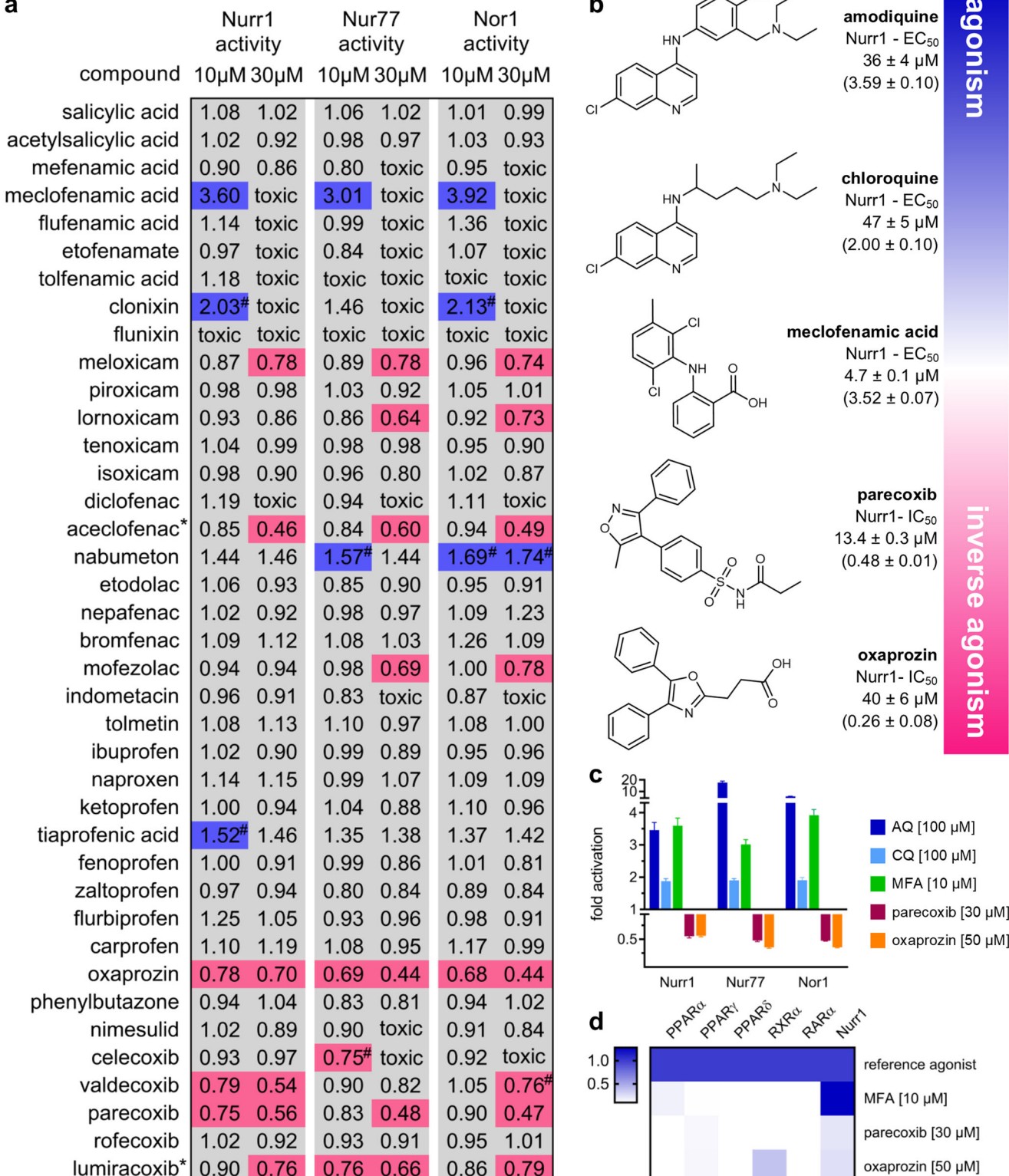

**Fig. 2 Bidirectional modulation of Nurr1 activity by drug-approved COX inhibitors. a** Screening of structurally diverse COX inhibitors for NR4A modulation in uniform Gal4-hybrid reporter gene assays. Heatmap shows mean fold activation; $n \geq 2$; agonists > 1.5-fold activation (blue), inverse agonists < 0.8-fold activation (magenta); all compounds displaying NR4A modulation in the primary screen were validated on Gal4-VP16 ($n \geq 4$) and only compounds not affecting Gal4-VP16 activity were further considered. Activities marked with # were not significant compared with VP16 control. Compounds marked with * were found to inhibit firefly luciferase. **b** Molecular structures and activities of Nurr1 modulators. AQ and CQ were reported as Nurr1 ligands previously[6]. EC$_{50}$ and IC$_{50}$ values were determined in the Gal4-Nurr1 hybrid reporter gene assay and are the mean ± SD; $n \geq 3$. Dose–response curves and control experiments on Gal4-VP16 hybrid receptor are shown in Fig. 3. **c** Activity profiles of Nurr1 modulators on the NR4A family receptors; mean fold activation ± S.E.M.; $n \geq 3$. **d** Selectivity profiles of MFA, parecoxib and oxaprozin on lipid-activated transcription factors outside the NR4A family. Heatmap shows mean rel. activation which refers to reference agonists at 1 μM for PPARs (α: GW7647; γ: rosiglitazone; δ: L165,041), RXRα (bexarotene), RARα (tretinoin) and 100 μM for Nurr1 (AQ); $n \geq 4$.

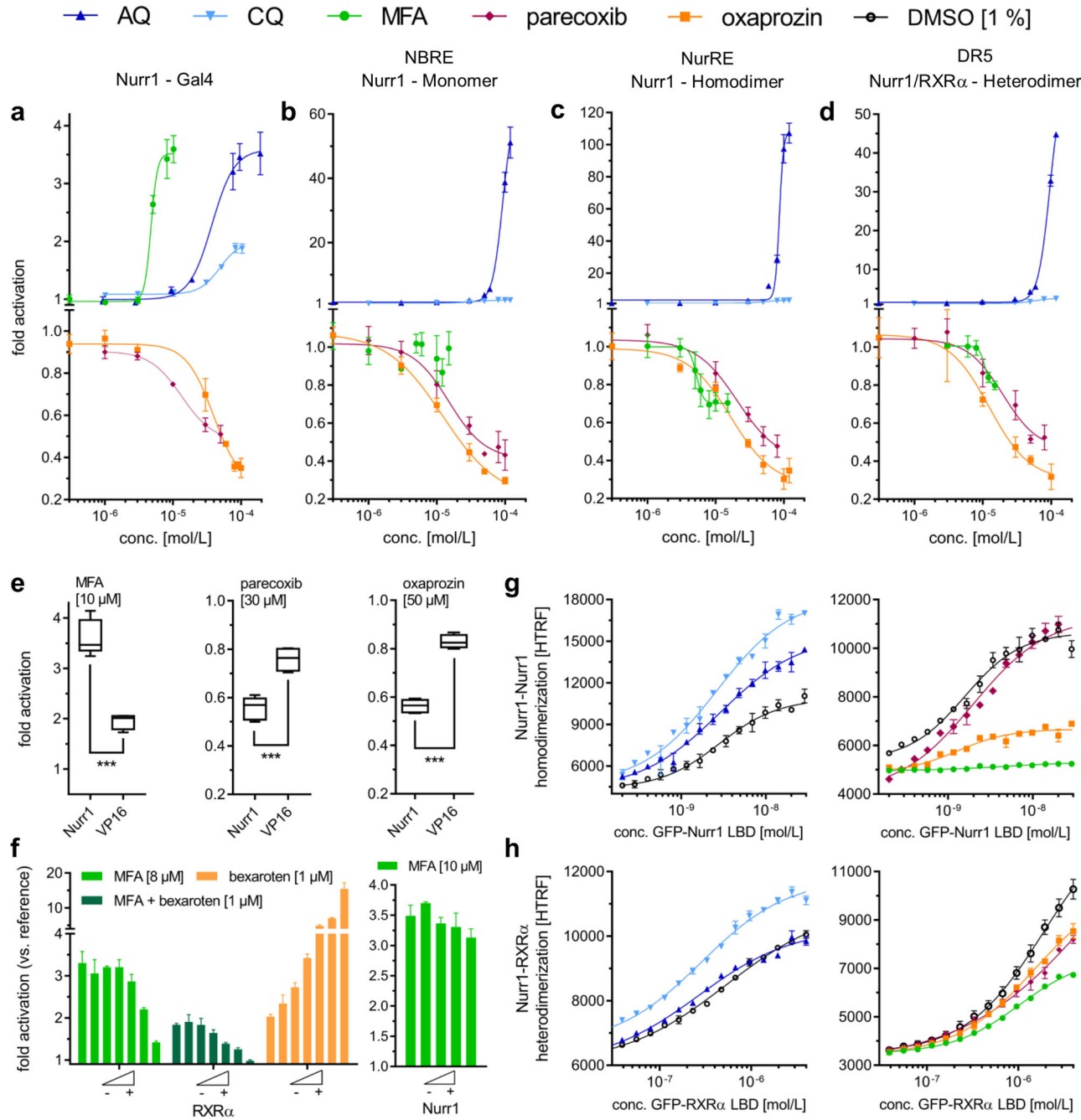

**Fig. 3 Cellular and cell-free profiling of Nurr1 modulation by small-molecule ligands. a** Gal4-hybrid reporter gene assay demonstrated Nurr1 activation by AQ, CQ, and MFA as well as inverse Nurr1 agonism for parecoxib and oxaprozin. **b–d** Nurr1 full-length reporter gene assays with the human Nurr1 response elements NBRE (Nurr1 monomer, **b**), NurRE (Nurr1 homodimer, **c**), and DR5 (Nurr1:RXR heterodimer, **d**) confirmed agonism of AQ and CQ (see also Supplementary Fig. 3) as well as inverse agonism of parecoxib and oxaprozin while MFA revealed a selective modulatory profile. All cellular experiments were performed in transiently transfected HEK293T cells. Results are the mean ± S.E.M.; $n \geq 3$. **e** Control experiments employing a Gal4-VP16 hybrid receptor confirmed Nurr1 mediated activity of MFA, parecoxib, and oxaprozin. Boxplots show: center line, median; box limits, upper and lower quartiles; whiskers, min/max; $n \geq 4$. *** $p < 0.001$. **f** With co-transfection of increasing amounts of Gal4-RXRα in the Gal4-Nurr1 reporter gene assay, activation efficacy of MFA (relative to DMSO) and MFA/bexarotene (relative to bexarotene) dropped pointing to monomer preference of MFA. Variations in the amount of Gal4-Nurr1 did not affect efficacy of MFA. Results are the mean ± S.E.M.; $n \geq 3$. **g** Nurr1 formed homodimers with high affinity in absence of ligands (DMSO). Nurr1 activators AQ and CQ promoted homodimerization. The inverse agonists parecoxib and oxaprozin diminished Nurr1 dimer formation and MFA entirely prevented homodimerization. Data are the mean ± S.E.M.; $N \geq 3$. **h** Nurr1 robustly heterodimerized with RXRα in apo state (DMSO). The Nurr1 activator CQ promoted dimerization between Nurr1 and RXRα whereas AQ was indifferent, and parecoxib, oxaprozin, and MFA had the opposite effect. Data are the mean ± S.E.M.; $N \geq 3$.

**Table 1 Summarized activities of Nurr1 modulators in cellular and cell-free experiments.**

| | AQ | CQ | MFA | Parecoxib | Oxaprozin |
|---|---|---|---|---|---|
| Gal4-Nurr1 | $EC_{50}$ 36 ± 4 (3.6 ± 0.1) | $EC_{50}$ 47 ± 5 (2.0 ± 0.1) | $EC_{50}$ 4.7 ± 0.1 (3.52 ± 0.05) | $IC_{50}$ 13.4 ± 0.3 (0.48 ± 0.01) | $IC_{50}$ 40 ± 6 (0.26 ± 0.08) |
| NBRE: Nurr1 monomer | $EC_{50}$ 92 ± 1 (62 ± 2) | $EC_{50}$ 38 ± 7 (2.2 ± 0.2) | Inactive | $IC_{50}$ 15 ± 3 (0.41 ± 0.05) | $IC_{50}$ 12 ± 2 (0.20 ± 0.05) |
| NurRE: Nurr1 homodimer | $EC_{50}$ 87 ± 2 (109 ± 5) | $EC_{50}$ 54 ± 7 (3.1 ± 0.2) | $IC_{50}$ 5.2 ± 0.1 (0.70 ± 0.01) | $IC_{50}$ 21 ± 6 (0.4 ± 0.1) | $IC_{50}$ 17 ± 3 (0.27 ± 0.05) |
| DR5: Nurr1-RXRα heterodimer | $EC_{50}$ 97 ± 3 (59 ± 4) | $EC_{50}$ 57 ± 8 (2.6 ± 0.2) | $IC_{50}$ 10.7 ± 0.1 (0.79 ± 0.01) | $IC_{50}$ 20 ± 8 (0.5 ± 0.1) | $IC_{50}$ 12 ± 2 (0.27 ± 0.05) |
| Homodimerization | Promotes dimerization | Promotes dimerization | Decreases dimerization | Inactive | Decreases dimerization |
| Heterodimerization | Inactive | Promotes dimerization | Decreases dimerization | Decreases dimerization | Decreases dimerization |
| NCoR-1 affinity | Enhances affinity | Enhances affinity | Decreases affinity | Decreases affinity | Inactive |
| NCoR ID1 recruitment | Inactive | $EC_{50}$ >100 | $IC_{50}$ 34 ± 3 | $IC_{50}$ 89 ± 8 | $IC_{50}$ >100 |
| SMRT ID2 recruitment | Inactive | $EC_{50}$ >100 | $IC_{50}$ 26 ± 4 | $IC_{50}$ 51 ± 5 | $IC_{50}$ 35 ± 5 |
| PRIPRAP250 recruitment | Inactive | $EC_{50}$ >100 | $IC_{50}$ 17 ± 2 | Inactive | $IC_{50}$ 72 ± 18 |
| RIP140L6 recruitment | Inactive | $EC_{50}$ 61 ± 18 | $IC_{50}$ 33 ± 3 | Inactive | $IC_{50}$ 104 ± 19 |

All $EC_{50}/IC_{50}$ values are shown in μM. Values in parentheses are maximum/minimum activation. All values are the mean ± SD.

valdecoxib in Supplementary Fig. 1d) outside the NR4A family, we studied their activity on other lipid- and fatty acid mimetic[13]-activated nuclear receptors (peroxisome proliferator-activated receptors, PPAR; retinoid X receptor α, RXRα; retinoic acid receptor α, RARα) which confirmed selectivity except weak RXR agonism of oxaprozin which has been described previously[14].

AQ, CQ, MFA, parecoxib, and oxaprozin displayed distinctive activity profiles of Nurr1 modulation ranging from agonism to inverse agonism, and therefore, emerged as a valuable set of tool compounds to assess Nurr1 modulation by chemical ligands.

**Modulation of Nurr1 depends on the DNA response element.** While the Gal4-hybrid reporter gene assay system is very reliable and provides a uniform setting for screening, it is also artificial. Physiologically, nuclear receptors have the ability to dimerize as a key regulatory interaction. They can act as monomers, homodimers or heterodimers with retinoid X receptor (RXR) the latter of which has also been suggested for Nurr1[15]. Understanding of Nurr1 modulation by ligands, thus, must also take ligand effects in more physiological settings into consideration where reporter activity is controlled by the native human full-length Nurr1 protein as monomer, homodimer, or RXR heterodimer. To study the effects of AQ, CQ, MFA, parecoxib, and oxaprozin on the activity of full-length human Nurr1 in cellular settings, we employed reporter constructs bearing a single repeat of the human DNA response elements for the Nurr1 monomer (NGFI-B response element, NBRE), the Nurr1 homodimer (Nur-response element, NurRE), or the Nurr1:RXR heterodimer (direct repeats spaced by 5 nucleotides, DR5) to control reporter gene expression. Nurr1 (and for DR5 also RXRα) was overexpressed by co-transfection of a CMV-dependent expression plasmid. These cellular assay settings revealed further differences for the individual Nurr1 modulators (Fig. 3b–d, Table 1). AQ robustly induced reporter activity on all response elements confirming Nurr1 agonism. CQ activated all Nurr1 reporters as well (Supplementary Fig. 3), but with markedly lower efficacy compared with AQ. Parecoxib and oxaprozin exhibited strong inverse agonism on NBRE, NurRE, and DR5 response elements, validating their inverse agonist activity, as well. MFA, however, revealed a more complex activity profile on the human Nurr1 response elements. On the monomer response element (NBRE), MFA was inactive while it suppressed activity of either Nurr1 dimer on NurRE and DR5 suggesting a selective Nurr1 modulatory profile.

**Nurr1 ligands modulate Nurr1 dimerization.** The observation of opposed effects of MFA on Nurr1 monomers and dimers suggested crucial involvement of Nurr1 dimerization in mediating responses to ligands. Therefore, we studied association of the LBDs of Nurr1 and RXR in time-resolved fluorescence resonance energy transfer (TR-FRET) based settings using GFP-labeled RXRα or Nurr1 LBDs and Tb-labeled Nurr1 LBD, from which the results demonstrated robust homodimeric (Fig. 3g) and heterodimeric binding (Fig. 3h) between the proteins in absence of a ligand. Consistent with our observations from the cellular settings, addition of Nurr1 ligands affected homo- and heterodimerization of Nurr1 in a distinctive fashion. The Nurr1 agonist AQ promoted formation of homodimers whereas the less effective agonist CQ enhanced homo- and heterodimerization. The inverse agonists parecoxib and oxaprozin, in contrast, diminished dimerization. Since oxaprozin also exhibits RXR agonism, its effects on heterodimerization must be interpreted with care, however. The Nurr1 modulator MFA exhibited the strongest effect on Nurr1 homodimerization and fully prevented formation of a Nurr1:Nurr1 dimer whereas heterodimerization was decreased in presence of MFA but not entirely disrupted.

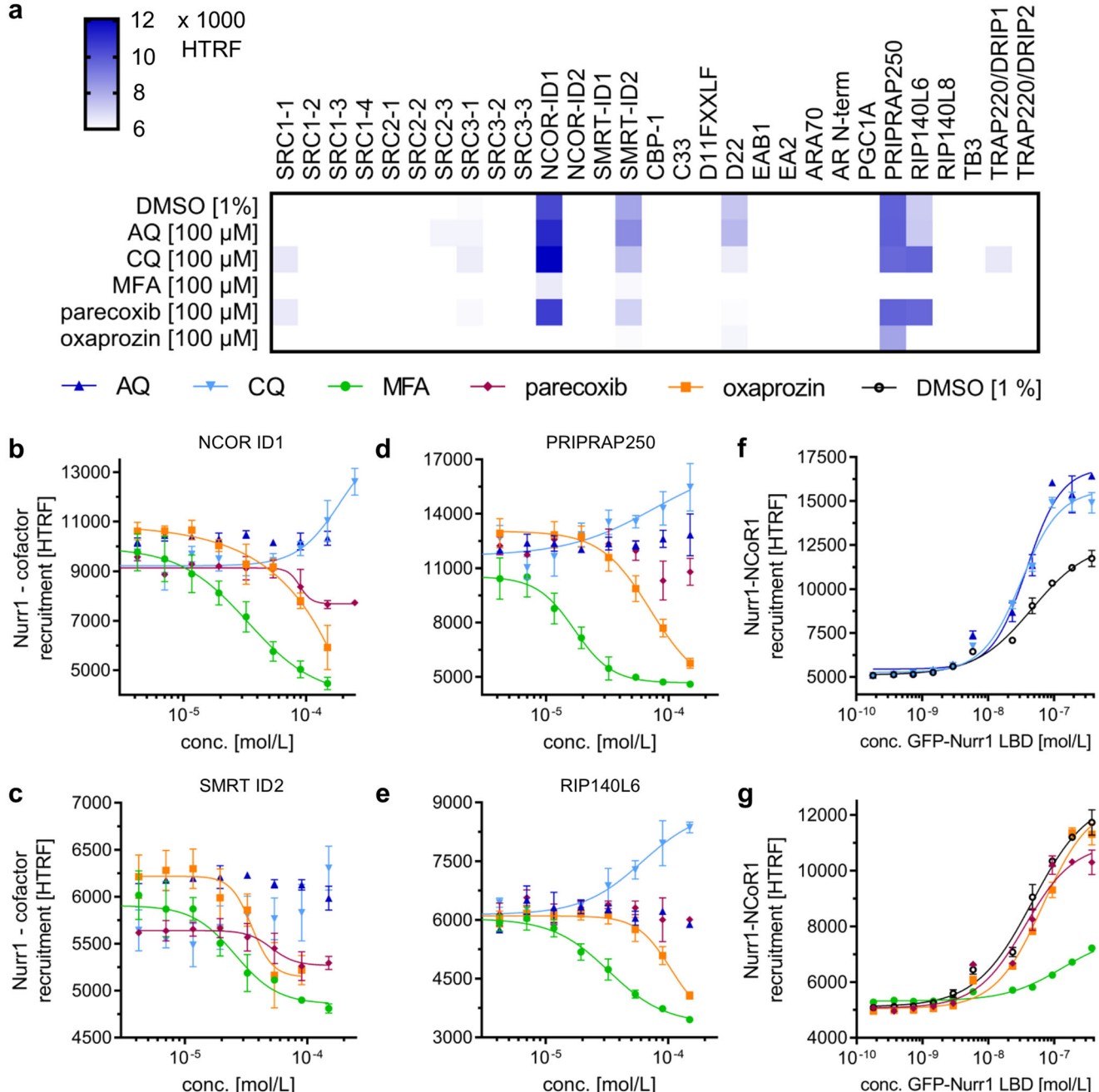

**Fig. 4 Interaction pattern of the Nurr1 LBD with co-regulators.** All interactions were studied in cell-free homogenous time-resolved fluorescence resonance energy transfer (HTRF)-based settings. Tb-labeled Nurr1 LBD as FRET donor and Fluorescein-labeled co-regulator peptides as FRET acceptors were used in (**a**–**e**). Tb-labeled NCoR-1 as FRET donor and GFP-labeled Nurr1 LBD as FRET acceptor were used in (**f**, **g**). **a** Twenty-nine peptides were screened for recruitment to Nurr1 in presence of 1% DMSO-control or ligands AQ, CQ, MFA, parecoxib, and oxaprozin at 100 μM. Heatmap of co-regulator recruitment screening shows the mean dimensionless HTRF signal, $N = 4$. **b**–**e** Dose–response curves of Nurr1 modulators in affecting recruitment of co-regulators NCoR-1 (**b**), SMRT (**c**), PRIPRAP250 (**d**), and RIP140 (**e**). Data are the mean ± S.E.M.; $N = 3$. **f**, **g** Binding curves for the Nurr1–NCoR-1 interaction in presence of 1% DMSO and AQ-type ligands (**f**) or NSAID-type ligands (**g**). Concentration of all ligands in (**f**) and (**g**) was fixed at 100 μM. Data are the mean ± S.E.M.; $N = 3$.

These observations further supported our hypothesis that the selective Nurr1 modulatory effects of MFA are mediated by changes in the dimerization state of the nuclear receptor. To observe this activity in another cellular setting, we studied how co-transfection of Gal4-RXRα affected MFA-mediated activation of Gal4-Nurr1 (Fig. 3f). In accordance with our previous results, increasing amounts of Gal4-RXRα resulted in a loss of activity of MFA while varying amounts of Gal4-Nurr1 had no effect.

Changes in the dimerization state of Nurr1, thus, emerge as key mechanism of Nurr1 modulation by small-molecule ligands. Therein, agonists (AQ, CQ) promote dimerization while inverse agonists (parecoxib, oxaprozin) diminish Nurr1 dimer formation.

**Nurr1 recruits canonical nuclear receptor co-regulators**. In addition to dimerization, nuclear receptor activity depends on interactions with various co-regulators. To capture also the

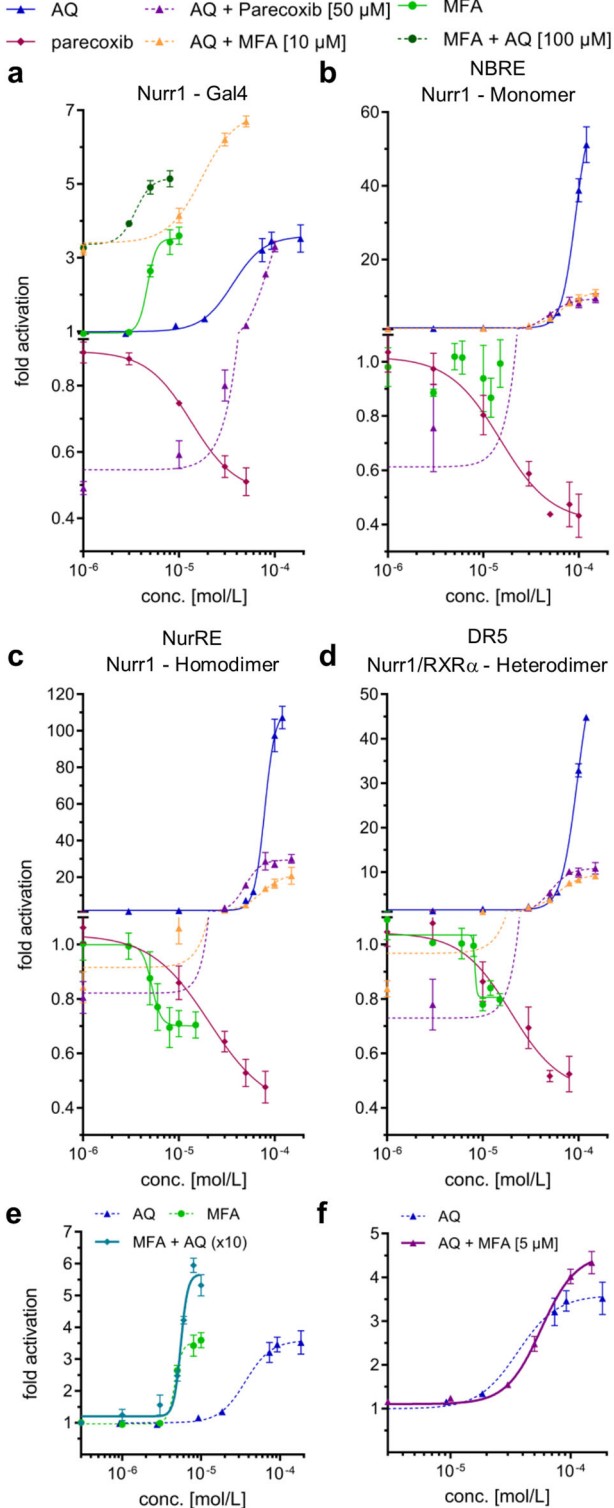

**Fig. 5 Simultaneous modulation of Nurr1 by AQ and NSAIDs in cellular reporter gene assays. a–d** Cross-titration curves of MFA or parecoxib and AQ on Gal4-Nurr1 (**a**) and on full-length human Nurr1 on the human Nurr1 response elements NBRE (**b**), NurRE (**c**), and DR5 (**d**). **e** Simultaneous titration of MFA and AQ (ratio 1:10) on Gal4-Nurr1. **f** Cross-titration experiment of AQ with lower fixed concentration of MFA (5 µM) in the Gal4-Nurr1 assay. All data are the mean ± S.E.M.; $n \geq 3$.

molecular mechanisms by which the Nurr1 co-regulator network responds to small-molecule ligands, we studied co-regulator binding to Nurr1 by TR-FRET in cell-free setting. We used a Tb-cryptate-labeled Nurr1 LBD and twenty nine Fluorescein-labeled co-regulator peptides (Fig. 4a). In absence of ligands, the Nurr1 LBD robustly recruited interaction motifs of nuclear receptor co-repressors 1 (NCoR-1) and 2 (NCoR-2, also termed silencing mediator for retinoid and thyroid hormone receptors, SMRT), nuclear receptor interacting protein 1 (NRIP1, also termed receptor interacting protein 140, RIP140), and nuclear receptor co-activator 6 (NCoA6, also termed TRBP, PRIP, RAP250). In agreement with earlier reports[16], no strong direct association of the Nurr1 LBD with the canonical steroid receptor co-activators (SRC) was observed.

**Nurr1 co-regulator interactions are responsive to ligands**. We then determined the effects of Nurr1 modulators AQ, CQ, MFA, parecoxib, and oxaprozin (all at 100 µM) on the recruitment of all twenty nine co-regulators to the Nurr1 LBD. While no pronounced effects were detected for AQ and CQ in this primary screen, MFA, parecoxib, and oxaprozin markedly altered the Nurr1 recruitment ability for NCoR-1, SMRT, PRIPRAP, and RIP140. In addition, the peptide D22 was recruited by Nurr1 in a ligand-dependent fashion but not further considered due to its artificial origin[17]. Full dose–response characterization further confirmed distinctive ligand effects on co-regulator recruitment. MFA, characterized as Nurr1 modulator in the cellular settings, displaced NCoR-1, SMRT, PRIPRAP, and RIP140 from the Nurr1 LBD in a dose-dependent fashion with similar potencies (IC$_{50}$ of 17–33 µM) (Table 1, Fig. 4b–e). The co-regulator recruitment profile of the inverse Nurr1 agonist oxaprozin resembled that of MFA despite lower potency of oxaprozin. Parecoxib, in contrast, only displaced NCoR-1 and SMRT from Nurr1 efficiently and merely tended to decrease the Nurr1–PRIPRAP and Nurr1–RIP140 interactions. The Nurr1 agonist CQ revealed a tendency to promote recruitment of NCoR-1, SMRT, PRIPRAP, and RIP140 while no effect was observed for AQ. However, due to the photophysical characteristics of AQ and CQ[18], and potential interference with the HTRF assay system through absorbance and quenching effects, these results for AQ and CQ must be interpreted with care. Thus, we employed a different setting and evaluated the affinity of Nurr1 for NCoR-1 binding in presence of the various ligands (Fig. 4f, g). Since ligand concentration was fixed in this setting, interference with the HTRF system is less prone to generate potential artifacts. Titration of GFP-labeled Nurr1 LBD against Tb-labeled NCoR-1 revealed markedly reduced affinity of Nurr1 for NCoR-1 recruitment in presence of MFA (100 µM) while AQ (100 µM) and CQ (100 µM) promoted the Nurr1–NCoR-1 interaction.

Thus, in addition to differential control of the Nurr1 dimerization state, ligands modulate Nurr1 activity by regulating cofactor recruitment. Agonists (CQ and potentially AQ) enhance recruitment of co-regulators such as NCoR-1, SMRT, PRIPRAP, and RIP140 while inverse agonists (parecoxib, oxaprozin) promote displacement of these co-factors.

**NSAIDs and amodiaquine simultaneously modulate Nurr1**. The PGA1 bound Nurr1 LBD X-ray structure (PDB-ID: 5Y41[7]) together with the mutagenesis and NMR-based analysis[6] of the putative binding site for AQ on Nurr1 suggest the existence of two independent ligand-binding pockets within the Nurr1 LBD (Fig. 1a) potentially allowing simultaneous modulation by small-molecule ligands. To test this hypothesis in vitro, we first treated

cells in the Gal4-Nurr1 reporter gene assay setting with either Nurr1 activator AQ or MFA at a fixed active concentration ($\geq$EC$_{90}$) and then monitored the activation upon cross-titrating the other respective agonist into the assay vice versa (Fig. 5a). Both compounds revealed additive effects and together (50 μM AQ and 10 μM MFA) achieved a strong Nurr1 activation, which clearly exceeded their individual activation efficacies (Fig. 5a, e, f). The EC$_{50}$ values of both compounds were not markedly affected by the presence of the other respective Nurr1 activator, suggesting that they interact with the receptor independently. Next, we performed a similar experiment but titrated both Nurr1 ligands in a fixed ratio of MFA/AQ 1:10 corresponding to their ~10-fold difference in potency, and observed a sigmoidal dose–response that reached considerably higher maximum activation efficacy than the individual compounds contradicting competitive behavior (Fig. 5e). Enhanced activation efficacy of Gal4-Nurr1 was also observed in cross-titration of AQ with lower fixed concentration (5 μM) of MFA (Fig. 5f). Furthermore, when we combined AQ with the inverse agonist parecoxib in cross-titration experiments (Fig. 5a), we found that the AQ dose–response curve was shifted to lower efficacy in presence of parecoxib. Together, these results strongly support our hypothesis of different binding sites for AQ and NSAIDs within the Nurr1 LBD.

We then expanded the cross-titration experiments to the more physiological settings of the full-length Nurr1 reporters (Fig. 5b–d) which agreed with our previous observations. On all three Nurr1 response elements, MFA (10 μM) and parecoxib (50 μM) prevented the full unfolding of AQ's agonistic potential even at high AQ concentrations suggesting simultaneous binding of either NSAID with AQ since with competitive antagonism, high AQ concentrations would displace the competitor and reach maximum efficacy.

## Discussion

The orphan nuclear receptor Nurr1 has been characterized as a neuroprotective and anti-neuroinflammatory transcription factor[19]. Evidence from animal models and human points to relevance of Nurr1 in PD[3,19,20], Alzheimer's disease[21,22], and multiple sclerosis[23–25] indicating a potential of the orphan nuclear receptor as therapeutic target in neurodegenerative diseases. However, the collection of Nurr1 modulators and knowledge on the receptor's molecular mode of action are limited advocating mechanistic studies on Nurr1 function and the search for new Nurr1 modulators as initial tool compounds for functional studies.

To assist validation of Nurr1 as future therapeutic target, there is a need for Nurr1 ligands as template for drug discovery and as chemical tools for biological studies to improve our knowledge on this orphan nuclear receptor. We have screened for alternative and additional Nurr1 modulators with higher potencies and distinct activity profiles, and employed them as in vitro tools for mode of action studies. Based on the recently published X-ray complex structure of the Nurr1 LBD bound to PGA1[7], which arises from cyclooxygenase activity, we hypothesized that COX inhibitors might potentially bind to Nurr1. We discovered the six NSAIDs MFA, aceclofenac, oxaprozin, valdecoxib, parecoxib, and meloxicam acting as Nurr1 modulators in cellular setting. Together with the previously reported[6] AQ-type Nurr1 ligands, the NSAID-type Nurr1 modulators discovered in our screening provide a valuable collection of initial tool compounds to evaluate Nurr1 activity covering activators and inverse agonists. These Nurr1 ligands demonstrate that the receptor's constitutive transcriptional inducer activity can be modulated by small molecules in a bidirectional fashion. Thereby, Nurr1 resembles other

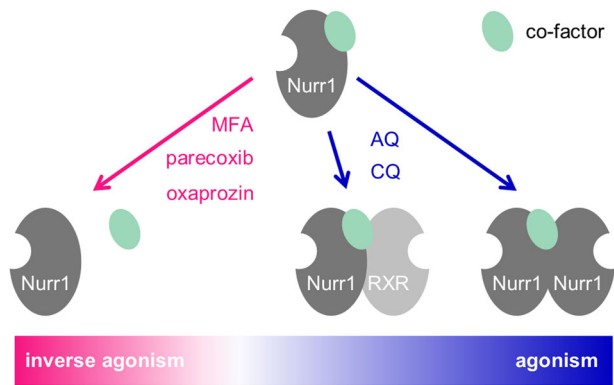

**Fig. 6 Model of Nurr1 responses to modulation by small-molecule ligands.** Agonists promote dimerization of Nurr1 as a homodimer or as a heterodimer with RXR and additionally stabilize interaction of the Nurr1 LBD with co-regulators. Inverse agonists favor monomeric Nurr1 and decrease Nurr1 co-regulator interactions.

nuclear receptors such as the RAR-related orphan receptors (RORs) which comprise high intrinsic activity and possess agonistic and inverse agonistic ligands[2]. Moreover, the simultaneous Nurr1 modulation by NSAIDs (MFA, parecoxib) and AQ observed in cross-titration experiments suggests potentially the existence of two ligand-binding pockets in Nurr1, both of which can control the receptor activity.

The Gal4-hybrid reporter gene assay employed for our primary screening is a robust test system but does not fully capture the physiological behavior of nuclear receptors in terms of dimerization. Jiang et al.[26] recently demonstrated that Nurr1 monomers, Nurr1 homodimers, and Nurr1:RXR heterodimers address distinct response elements on DNA. To transfer our findings to endogenous conditions with different Nurr1 DNA response elements and dimerization states, we profiled the entire set of initial Nurr1 tool compounds (AQ, CQ, MFA, parecoxib, oxaprozin) in reporter gene assays involving the human Nurr1 response elements NBRE, NurRE, and DR5 as well as the full-length human Nurr1 protein as monomer, homodimer, or heterodimer. Dose–response experiments on NBRE-, NurRE-, and DR5-dependent reporter expression confirmed agonism of AQ and CQ as well as inverse agonism of parecoxib and oxaprozin on Nurr1 monomers (NBRE), homodimers (NurRE), and heterodimers (DR5). MFA, however, exhibited a less consistent activity profile with agonism on Gal4-Nurr1, inverse agonism on Nurr1 dimers (NurRE, DR5) and no activity on full-length Nurr1 monomers (NBRE). This observation suggests promising potential for (gene-)selective modulation of Nurr1 but also demonstrates limitations of the Gal4-hybrid assay setting to study Nurr1 modulation since it was not predictive of MFA's mode of activity on full-length human Nurr1.

In order to observe the molecular modes by which agonists and inverse agonists differentially modulate Nurr1, we studied the dimerization behavior of the Nurr1 LBD and its interaction with co-regulators in presence and absence of ligands. We observed high affinity of Nurr1 to homodimerize and to form heterodimers with RXR. These interactions were markedly affected by Nurr1 modulators and enable distinction between agonism and inverse agonism with agonists promoting dimer formation and inverse agonists decreasing dimerization. MFA was most effective in countering Nurr1 dimer formation regarding both, homo- and heterodimer which suggests that MFA shifts the binding equilibrium of Nurr1 to a fully monomeric state. This is in line with different behavior of the compound in cellular settings involving monomers (Gal4-assay, NBRE) or dimers (NurRE, DR5) and

provides preliminary explanation why the observed activities of MFA differed in different cellular settings. In addition, the ligand-induced Nurr1 dimerization state appears to affect Nurr1 activation efficacy since AQ which primarily promoted Nurr1 homodimerization in cell-free setting concomitantly exhibited the strongest activation efficacy on the human Nurr1 homodimer response element (NurRE, >100-fold max. activation).

We then assessed how ligands modulate the interaction between Nurr1 and co-regulators, and screened a library of twenty nine known nuclear receptor co-regulator fragments for their recruitment to Nurr1. We observed robust binding of NCoR-1, NCoR-2, NCoA6, and NRIP1 to the Nurr1 LBD and discovered that these interactions are responsive to Nurr1 ligands. The Nurr1 agonist CQ revealed a trend to enhanced recruitment of co-regulators and both AQ and CQ increased the affinity between the Nurr1 LBD and NCoR-1 while inverse agonists parecoxib and oxaprozin displaced co-regulator interaction motifs from the Nurr1 LBD. Different effects of Nurr1 modulators on co-regulator recruitment to the Nurr1 LBD therefore arise as further contributing factor to discriminate Nurr1 agonism and inverse agonism. In addition, different effects of AQ and CQ on co-regulator recruitment by Nurr1—beyond their distinctive modulation of Nurr1 dimerization—provide a basis for their different Nurr1 activation efficacies. Using a limited coverage of potential interactors, our results indicated strongly a similar ability of Nurr1 to other nuclear receptors for interactions with a number of co-regulators and we postulate that further biologically-relevant co-regulators might involve in Nurr1 regulation.

Based on our findings in orthogonal cellular and cell-free assay systems, we conclude that bidirectional modulation of Nurr1 by small-molecule ligands results from two contributions. Agonists promote dimerization of Nurr1 as a homodimer or as a heterodimer with RXR and additionally stabilize interaction of the Nurr1 LBD with co-regulators. Inverse agonists favor monomeric Nurr1 and decrease Nurr1 co-regulator interactions (Fig. 6). These changes in protein–protein interactions of Nurr1 result in differential cellular effects on reporter/gene expression. Therein, MFA exhibited a profile between agonism and inverse agonism on different Nurr1 response elements and since these DNA motifs are found in the promoter regions of different genes, gene selective modulation of Nurr1 activity seems possible with ligands that either favor monomeric or dimeric states. These considerations give our observation of ligand-dependent preference of Nurr1 for monomeric or dimeric forms considerable relevance.

Overall our studies have demonstrated that Nurr1 activity can be modulated by small-molecule ligands in a bidirectional fashion and we report first-in-class inverse Nurr1 agonists that counter the receptor's high constitutive transcriptional inducer activity. The opposite activities of Nurr1 agonists and inverse agonists are rationalized by their distinct effects on Nurr1's interaction profile with co-regulators and dimerization state. Moreover, our results point to the existence of two binding sites within the Nurr1 LBD that can accommodate ligands, of which the binding can modulate Nurr1 activity both independently and simultaneously. This is demonstrated by pairs of modulators involving AQ and MFA or parecoxib, which achieve different Nurr1 activation efficacy when they are applied together compared with their individual effects, offering potentially another avenue for the design of selective Nurr1 modulators or agonists with enhanced efficacy. These results markedly contribute to molecular understanding of Nurr1's activity and advocate the development of several types of potent Nurr1 modulator tool compounds that address the different binding sites and distinct receptor responses to enable in depth functional validation of Nurr1 as future drug target.

## Methods

**Reporter gene assays**. *Plasmids*: The Gal4-fusion receptor plasmids pFA-CMV-hNur77-LBD, pFA-CMV-hNURR1-LBD, and pFA-CMV-hNOR1-LBD coding for the hinge region and LBD of the canonical isoforms of the human nuclear receptors Nur77 (uniprot entry: hNUR77–P22736, residues 358–598), Nurr1 (uniprot entry: hNURR1–P43354, residues 360–598), and NOR1 (isoform alpha; uniprot entry: hNOR1–Q92570-1, residues 393–626) were constructed by integrating cDNA fragments obtained from PCR amplification using natural cDNA (Nur77: GenBank entry: BC016147.1, purchased as I.M.A.G.E. cDNA clone from Source BioScience, Nottingham, UK; Nurr1: GenBank entry: BC009288.2, purchased as I.M.A.G.E. cDNA clone from Source BioSience) or the pcDNA3.1 plasmid OHu22293D (NOR1; GenScript, USA; NCBI ref. NM_173200.2) as template between the BamHI cleavage site of the pFA-CMV vector (Stratagene, La Jolla, CA, USA) and an afore inserted KpnI cleavage site. Frame and sequence of the fusion plasmids were verified by sequencing. The Gal4-fusion receptor plasmids used for selectivity profiling were pFA-CMV-hPPARα-LBD, pFA-CMV-hPPARγ-LBD, pFA-CMV-hPPARδ-LBD, pFA-CMV-hRXRα-LBD, and pFA-CMV-hRARα-LBD coding for the hinge region and ligand-binding domain of the canonical isoform of the respective nuclear receptor have been reported previously[27–29]. pFR-Luc (Stratagene) was used as reporter plasmid and pRL-SV40 (Promega, Madison, WI, USA) for normalization of transfection efficiency and test compound toxicity. The Gal4-VP16[12] expressed from plasmid pECE-SV40-Gal4-VP16[30] (Addgene, entry 71728, Watertown, MA, USA) was used as ligand-independent transcriptional inducer for control experiments. The reporter plasmid pFR-Luc (Stratagene) used for the Gal4-hybrid assays contains a section between 176 to 83 base pairs upstream of the start codon of the firefly CDS that encompasses five copies of the Gal4 response element. To enable transactivation assays based on full-length NRs, this section was replaced with the human Nurr1 response elements DR5 (pFR-Luc-DR5; TGATAGGTTCACCGAAAGGTCA), NBRE NL3 (pFR-Luc-NBRE; TGA-TATCGAAAA̲CAAAAGGTCA), or NurRE (from proopiomelanocortin (POMC); pFR-Luc-NurRE; T̲G̲A̲T̲A̲T̲T̲T̲A̲CCTCCAAATGCCA), respectively. The human nuclear receptors Nurr1 (pcDNA3.1-hNurr1-NE; #102363, Addgene, Cambridge, MA, USA) and, for DR5, RXRα (pSG5-hRXR[31]) were overexpressed. *Assay procedure*: HEK293T cells (German Collection of Microorganisms and Cell Cultures (DSMZ), Braunschweig, Germany) were grown in DMEM high glucose, supplemented with 10% FCS, sodium pyruvate (1 mM), penicillin (100 U/mL), and streptomycin (100 μg/mL) at 37 °C and 5% CO₂. The day before transfection, HEK293T cells were seeded in 96-well plates ($3 \times 10^4$ cells/well). Before transfection, medium was changed to Opti-MEM without supplements. Transient transfection was performed using Lipofectamine LTX reagent (Invitrogen, Carlsbad, CA, USA) according to the manufacturer's protocol with the corresponding plasmid mixture. For Gal4-hybrid assays, the plasmid mixtures comprised the respective Gal4-fusion nuclear receptor plasmid (pFA-CMV-NR-LBD), pFR-Luc, and pRL-SV40. For assays on full-length human Nurr1, the plasmid mixtures were pcDNA3.1-hNurr1-NE/pFR-Luc-NBRE/pRL-SV40 (NBRE), pcDNA3.1-hNurr1-NE/pFR-Luc-NurRE/pRL-SV40 (NurRE), and pcDNA3.1-hNurr1-NE/pSG5-RXR/pFR-Luc-DR5/pRL-SV40 (DR5). Five hours after transfection, medium was changed to Opti-MEM supplemented with penicillin (100 U/mL) and streptomycin (100 μg/mL), now additionally containing 0.1% DMSO and the respective test compound or 0.1% DMSO alone as untreated control. Each concentration was tested in duplicates and each experiment was performed independently at least three times. Following overnight (12–14 h) incubation with the test compounds, cells were assayed for luciferase activity using Dual-Glo™ Luciferase Assay System (Promega) according to the manufacturer's protocol. Luminescence was measured with a Spark 10 M luminometer (Tecan Group Ltd., Männedorf, Switzerland). Normalization of transfection efficiency and cell growth was done by division of firefly luciferase data by renilla luciferase data and multiplying the value by 1000 resulting in relative light units (RLU). Fold activation was obtained by dividing the mean RLU of a test compound at a respective concentration by the mean RLU of untreated control. Max. relative activation refers to fold reporter activation of a test compound divided by the fold activation of the respective reference agonist (PPARα: GW7647; PPARγ: pioglitazone/rosiglitazone[29]; PPARδ: L165,041; RXRα: bexarotene; RARα: tretinoin; all at a concentration of 1 μM; Nurr1: amodiaquine (100 μM)). All hybrid assays were validated with the above mentioned reference agonists which yielded $EC_{50}$ values in agreement with the literature.

**Production of recombinant RXRα and Nurr1 fusion proteins**. The coding sequence for RXRα LBD and Nurr1 LBD was codon optimized for *E. coli* and purchased from Geneart (Regensburg, Germany), respectively. For expression of fusion proteins with N-terminal green fluorescent protein (GFP), an expression construct based on pET29b was prepared. For this, the entire section between the original NdeI site and the forth position following the His-Tag coding sequence of pET29b was replaced, hence, essentially leaving only the vector backbone unmodified. The section was replaced by a sequence encoding a restriction site for NcoI (overlapping with the start codon) and an open reading frame for Met-Gly-[His₁₀-Tag]-Asp-Tyr-Asp-Ile-Pro-Thr-Thr-[TEV site]-superfolder GFP[32] followed by restriction sites for BamHI (in frame) and XhoI. The sequences coding for the LBDs of RXRα (uniprot entry: P19793-1, residues 226–462) or Nurr1 (uniprot entry: P43354-1, residues 364–598) each followed by a stop codon were then

introduced in frame between the afore inserted restriction sites for BamHI and XhoI.

For generation of biotinylated Nurr1 LBD, the pMal vector system (New England Biolabs, NEB, Ipswich, MA, USA) was used. In pMal-c2E, the section between the sequence encoding 10x Asparagine ($Asn_{10}$) and the SalI restriction site was replaced with a sequence encoding Leu-Gly-Ile-Glu-Leu-Val-[$His_8$-Tag]-Asp-Tyr-Asp-Ile-Pro-Gly-Thr-Leu-[TEV site] followed by an Avi-Tag and restriction sites for BamHI and XhoI. The sequence encoding Nurr1 (aa 364–598) followed by two stop codons was cloned in frame between these restriction sites. From this construct, a fusion protein is expressed with N-terminal maltose-binding protein (MBP) followed by an $Asn_{10}$ linker, a $His_8$-Tag, a cleavage site for TEV protease, an Avi-Tag, and the Nurr1 LBD with unmodified C-terminus.

For expression, *E. coli* T7 express cells (NEB) were co-transformed with pGro7 (TAKARA Bio Inc., Kusatsu, Japan) and one of the Nurr1 (pMal or pET) or RXRα (pET) expression constructs and selected overnight at 37 °C on LB (Luria Broth) agar containing 34 μg/ml chloramphenicol and either 100 μg/ml ampicillin (for pMal) or 35 μg/ml kanamycin (for pET). Culture in liquid LB was inoculated and grown at 37 °C with constant shaking at 180 rpm until optical density at 600 nm ($OD_{600}$) reached 0.7. At this time point, expression of the chaperone GroEL/ES from pGro7 was induced with 1 g/L L(+)-Arabinose and the temperature was reduced to 20 °C. At $OD_{600} = 1$ expression of the target protein was induced by addition of 0.5 mM IPTG. After 12–16 h, cells were harvested by centrifugation and resuspended in buffer A (400 mM NaCl, 20 mM $NaP_i$ pH 7.8, 10% (w/v) Glycerol, and 20 mM ß-mercaptoethanol). Cells were kept on ice and disrupted in presence of 1 mM ATP, DNAse I, RNAse A, 20 mM $MgSO_4$, and EDTA-free cOmplete™ protease inhibitor cocktail (F. Hoffmann-La Roche AG, Basel, Switzerland) by addition of lysozyme and 10 passages through an Invensys APV-1000 homogenizer (APV Systems, Silkeborg, Denmark). Cell debris was removed by centrifugation at $16,500 \times g$ for 20 min at 4 °C.

Purification was achieved by immobilized metal chromatography (IMAC) using columns packed with Ni Sepharose 6 Fast Flow resin on an ÄKTApurifier FPLC system (GE Healthcare, Chicago, IL, USA). After washing with buffer supplemented with 50 mM imidazole the protein was eluted with 300 mM imidazole. Afterward, GFP fusion proteins were processed with His tagged TEV protease overnight while imidazole content was reduced to 10 mM by dialysis against buffer A in order to allow for reverse IMAC. The flow through was concentrated and applied to size exclusion chromatography using a 16/60 Superdex200™ column equilibrated and run in HTRF assay buffer [25 mM HEPES pH 7.5, 150 mM KF, 10% (w/v) glycerol, 5 mM DTT]. Following the initial IMAC purification step, the MBP fusion protein for generation of biotin-labeled Nurr1 LBD was processed with MBP-tagged TEV protease during overnight dialysis against buffer A. Afterward, uncleaved fusion protein, free MBP-Tag, and TEV protease were removed by passaging through a gravity flow column packed with Amylose High Flow resin (NEB). The flow through was then supplemented with 0.5 mM biotin, 0.5 mM ATP, 5 mM $MgCl_2$, and *E. coli* biotin ligase birA at a molar ratio of ~1:10 for enzymatic conjugation of biotin to the lysine residue in the avitag. After overnight incubation at 4 °C, the solution was subjected to a column packed with 5 ml monomeric avidin UltraLink™ resin (Pierce Biotechnology Inc., Rockford, IL, USA). Unlabeled protein and birA were removed by washing for 10 column volumes with buffer A before biotin-labeled Nurr1 LBD was eluted using buffer A supplemented with 2 mM biotin. The product was then concentrated and subjected to size exclusion chromatography using a 10/30 Superdex75™ column equilibrated and run in HTRF assay buffer.

**Nurr1 co-regulator recruitment assays**. Recruitment of co-regulator peptides to the Nurr1 LBD was studied in a homogeneous time-resolved fluorescence resonance energy transfer (HT-FRET) assay system. Terbium cryptate as streptavidin conjugate (Tb-SA; Cisbio Bioassays, Codolet, France) was used as FRET donor for stable coupling to biotinylated recombinant Nurr1-LBD protein. Twenty-nine co-regulator peptides fused to fluorescein as FRET acceptor were purchased from ThermoFisher Scientific (Life Technologies GmbH, Darmstadt, Germany). Assay solutions were prepared in HTRF assay buffer supplemented with 0.1% (w/v) CHAPS and contained recombinant biotinylated Nurr1 LBD (final concentration 3 nM), Tb-SA (3 nM), and the respective fluorescein-labeled co-regulator peptide (100 nM) as well as 1% DMSO with test compounds at 100 μM or DMSO alone as negative control. All HTRF experiments were carried out in 384 well format using white flat bottom polystyrol microtiter plates (Greiner Bio-One, Frickenhausen, Germany). After 2 h incubation at RT, fluorescence intensities (FI) after excitation at 340 nm were recorded at 520 nm for fluorescein acceptor fluorescence and 620 nm for Tb-SA donor fluorescence on a SPARK plate reader (Tecan Group Ltd.). FI520nm was divided by FI620nm and multiplied with 10,000 to give a dimensionless HTRF signal. Dose–response experiments with varying concentrations of the test compounds amodiaquine, chloroquine, meclofenamic acid, parecoxib, and oxaprozin were conducted in the same manner and setting. The co-regulator peptides in this experiment were the following: steroid receptor co-activator (SRC) 1-1, Fluorescein-KYSQTSHKLVQLLTTTAEQQL-OH; SRC 1-2, Fluorescein-LTARHKILHRLLQEGSPSD-OH; SRC 1-3, Fluorescein-ESKDHQLLRYLLDK-DEKDL-OH; SRC 1-4, Fluorescein-GPQTPQAQQKSLLQQLLTE-OH; SRC 2-1, Fluorescein-DSKGQTKLLQLLTTKSDQM-OH; SRC 2-2, Fluorescein-LKEKH KILHRLLQDSSSPV-OH; SRC 2-3, Fluorescein-KKKENALLRYLLDKDDTKD-

OH; SRC 3-1, Fluorescein-ESKGHKKLLQLLTCSSDDR-OH; SRC 3-2, Fluorescein-LQEKHRILHKLLQNGNSPA-OH; SRC 3-3, Fluorescein-KKENNALLR YLLDRDDPSD-OH; nuclear receptor co-repressor (NCOR) ID1, Fluorescein-RTHRLITLADHICQIITQDFARN-OH; NCOR ID2, Fluorescein-DPASNLGLE DIIRKALMGSFDDK-OH; silencing mediator for retinoid and thyroid hormone receptor (SMRT) ID1, Fluorescein-GHQRVVTLAQHISEVITQDYTRH-OH; SMRT ID2, Fluorescein-HASTNMGLEAIIRKALMGKYDQW-OH; CREB-binding protein 1 (CBP-1), Fluorescein-AASKHKQLSELLRGGSGSS-OH; C33, Fluorescein-HVEMHPLLMGLLMESQWGA-OH; D11-FXXLF, Fluorescein-VESGSSRFMQLFMANDLLT-OH; D22, Fluorescein-LPYEGSLLLKLLRAPVEEV-OH; EAB1, Fluorescein-SSNHQSSRLIELLSR-OH; EA2, Fluorescein-SSKGV LWRMLAEPVSR-OH; androgen receptor-associated protein 70 (ARA70), Fluorescein-SRETSEKFKLLFQSYNVND-OH; N-terminal sequence of androgen receptor (AR N-term), Fluorescein-SKTYRGAFQNLFQSVREVI-OH; peroxisome proliferator-activated receptor gamma co-activator 1-alpha (PGC1a), Fluorescein-EAEEPSLLKKLLLAPANTQ-OH; nuclear receptor co-activator 6 (NCoA6, also termed PRIPRAP250), Fluorescein-VTLTSPLLVNLLQSDISAG-OH, nuclear receptor interacting protein 1 (NRIP1, also termed RIP140, interaction motif L6), Fluorescein-SHQKVTLLQLLLGHKNEEN-OH; RIP140L8, Fluorescein-SFSKNGLLSRLLRQNQDSY-OH; TB3, Fluorescein-SSVASREWWVRELSR-OH; thyroid hormone receptor-associated protein (TRAP) TRAP220/DRIP-1, Fluorescein-KVSQNPILTSLLQITGNGG-OH; TRAP220/DRIP-2, Fluorescein-NTKNHPMLMNLLKDNPAQD-OH.

**Nurr1–RXR heterodimerization**. Strength and modulation of the formation of the heterodimer composed of the LBDs of Nurr1 and RXRα was investigated by titration of GFP-RXRα LBD against a fixed concentration of Nurr1 LBD. Assay solutions were prepared in HTRF assay buffer supplemented with 0.1% (w/v) CHAPS as well as 1% DMSO with test compounds at 100 μM or DMSO alone as negative control. The FRET donor complex formed from biotinylated Nurr1 LBD (final concentration 0.375 nM) and Tb-SA (0.75 nM) was kept constant while the concentration of GFP-RXRα LBD was varied starting with 4 μM as the highest concentration and titrated with a dilution factor of 0.7. Free GFP was added to keep the total GFP content stable at 4 μM throughout the entire series in order to suppress artefacts from changes in degree of diffusion enhanced FRET. Samples were equilibrated at RT for 2 h before FI520 and FI620 were recorded after excitation at 340 nm, and the HTRF signal was calculated as described above.

**Nurr1 homodimerization**. Nurr1 homodimerization was studied by the same strategy using GFP-Nurr1 LBD instead of GFP-RXRα LBD. Since affinity observed for Nurr1 homodimer formation was higher, the maximum concentration for GFP-Nurr1 LBD and the total GFP concentration was reduced to 500 nM.

**Nurr1–NCoR-1 interaction**. The Nurr1–NCoR-1 interaction was studied by titrating GFP-Nurr1 LBD against biotinylated NCoR-1 copeptide (18 nM) and Tb-SA (12 nM) in presence of a fixed concentration (100 μM, in assay buffer containing 1% DMSO) of the respective ligand or 1% DMSO. To maintain a constant GFP concentration, free GFP protein was added to the dilution series. The experiments were performed in HTRF assay buffer (150 mM KF, 25 mM HEPES pH 7.5 (KOH), 5% (w/v) Glycerol, supplemented with 0.1% (w/v) CHAPS and 5 mM DTT) with 1% DMSO in an assay volume of 20 μl. After 1 h incubation at RT, fluorescence intensities after excitation at 340 nm were recorded at 520 nm for GFP acceptor fluorescence and 620 nm for Tb-SA donor fluorescence and the HTRF signal was calculated as described above.

**Computational methods**. *General*: Calculations were conducted in Molecular Operating Environment (MOE, version 2018.0101, Chemical Computing Group Inc., Montreal, QC, Canada) using default settings for each tool/function unless stated otherwise. *Crystal structure analysis*: Alignment of the Nurr1 LBD in apo state (PDB: 1OVL[1]), bound to prostaglandin A1 (PDB: 5Y41[7]) and to dopamine metabolite DHI (PDB: 6DDA[9]) was conducted using MOE sequence editor. The subunits B were used in all cases. The proposed binding region for amodiaquine type ligands was highlighted according to annotated amino acids from NMR perturbation experiments and mutational studies[6]. Distances of the salt bridge between Lys590 to Glu440 were measured in angstrom (Å) in MOE from nitrogen ($NH_3^+$) of Lys590 to oxygen ($O^-$) of Glu440. Data shown are the mean ± SD corresponding to the different subunits of the X-ray structures.

**Statistics**. Calculations and graphical analysis of experimental data was conducted using GraphPad Prism version 7.00 for Windows (GraphPad Software, La Jolla, CA, USA) and Microsoft Excel 2016 (Microsoft Corporation, Redmond, WA, USA). All cellular experiments were performed with at least three independent biological repeats ($n \geq 3$), each in duplicates. The cell-free experiments were performed with three technical replicates ($N = 3$), whereas the cofactor screen was performed with four technical replicates ($N = 4$). All dose–response curves were calculated in GraphPad Prism using a nonlinear regression with variable slope ([Agonist] or [Inhibitor] vs. response; four parameters). Statistical significance was evaluated by two-tailed student's *t*-test (two samples, unequal variance; calculated in Excel) with $n \geq 4$. Results were considered statistically significant with

$p$ values < 0.05; significance levels are denoted as *$p$ < 0.05, **$p$ < 0.01, ***$p$ < 0.001. Boxplots were generated in GraphPad Prism and show: center line, median; box limits, upper and lower quartiles; whiskers, min/max; $n \geq 4$.

**Reporting summary**. Further information on research design is available in the Nature Research Reporting Summary linked to this article.

## Data availability

The datasets generated and analyzed during the current study are available from the corresponding author on reasonable request.

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

## Acknowledgements

D.M. is grateful for support by the Aventis Foundation. This work was supported by the research funding program LOEWE of the State of Hessen, Research Center for Trans-lational Medicine and Pharmacology TMP.

## Author contributions

S.W., W.K., J.H. and D.M. performed the experiments. S.W. and D.M. analyzed the Nurr1 LBD structures. X.N., A.C. and S.K. generated recombinant Nurr1 LBD protein. J.H. cloned the NR4A receptor and full-length reporter constructs. D.M. designed and supervised the study. S.W. and D.M. analyzed the data, prepared the figures, and wrote the paper.

## Competing interests

The authors declare no competing interests.
