## [Peer Review File · Communications Chemistry]

Reviewers' comments:

Reviewer #1 (Remarks to the Author):

Willems et al. describe the characterization of a series of structurally diverse non-steroidal anti-inflammatory drugs as Nurr1 modulators. In the context of a Gal4-Nurr1 LBD construct, they act as agonists or inverse agonists, and biochemical studies confirm that they modulate the binding of a series of corepressor peptides to the Nurr1 LBD. These studies are interesting as far as they go but fall well short of demonstrating that the compounds are indeed agonists and inverse agonists. Three important further characterizations are needed.

The first is simply (and remarkably) that the effect of the compounds on the intact Nurr1 receptor has not been assessed. This could be done easily by cotransfection with a standard NBRE reporter but should also include the dimeric ER0 and IR5 response elements noted in the discussion. To more critically test their effects on the native Nurr1, the response of a range of endogenous Nurr1 target genes should also be defined, with and without Nurr1 knockdown.

The second is that they have not tested whether the compounds affect the activity of the other 2 NR4A family members. This is critical for assessment of their potential biologic activity.

Finally, this story could be significantly strengthened by using mutagenesis to probe the NSAID ligand binding site. The clearly demonstrated additive effects with amodiaquine strongly suggest that the other identified pocket mediates NSAID compound effects, but that should be tested experimentally. Ideally, the mutagenesis studies could be done in the presence and absence of amodiaquine to functionally define the distinct sites.

Reviewer #2 (Remarks to the Author):

Given its decreased expression in dopaminergic neurons, NURR1 appears to be an excellent target for disease modulation like Parkinson's disease. Despite initial crystal structures that did not define a clear ligand binding pocket, indicating its transcriptional activity may be ligand-independent, recent reports have suggested it may indeed be ligand-regulated. Despite this, few potent NURR1 modulators have been described. Based on a recent co-crystal structure of NURR1 bound to prostaglandin A1, Willems et al. hypothesized that COX inhibitors may be NURR1 ligands. After a screen of COX-1 and COX-2 inhibitors, they identified several structurally distinct ligands that appear to function as NURR1 agonists and inverse agonists in vitro using a Gal4 NURR1 transactivation assay. These data support the co-crystal structure data and identify COX inhibitors as potential therapeutics to exploit for further drug design.

The concepts presented in the manuscript are interesting and support previously published work indicating that PGA2 regulates NR4A2. So that fact that COX inhibitors modulate NR4A2 is in-line with this. However, there are several points that need addressing to strengthen the conclusions:

1) Figure 2 could be strengthened by performing knock downs of NURR1 in the cell-based assays to prove that the compounds are NURR1 selective. Furthermore, it would be ideal to screen these compounds against other nuclear receptors to determine how selective these ligands are, particularly against members of the NR4A family (NURR77 and NOR1) and other nuclear receptor subfamilies with members demonstrated to be modulated by prostaglandins – the PPARs.

2) Figure 3 is hardly indicative of the notion that the 2 ligands (meclofenamic acid and amodiaquine) are interacting with the receptor independently. Further studies need to be performed to even possibly make this claim: a) perform the transactivation assays but titrate the

ligands down to significantly lower concentrations so one can see the actual intersection of the 2 ligands when they are not activating the receptor. b) perform a biochemical based study to prove the 2 binding events like isothermal calorimetry or CD thermal melt. Given that cloroquine is also a reported ligand for NURR1, similar studies should be performed to validate the effects observed with amodiaquine.

3) The fact that the amodiaquine had "no effects" due to squelching is highly unlikely given that a) it did recruit SRC2 and SRC3 peptides (albeit weakly) at 100uM, which is consistent with previous reports for SRC3 and b) one would expect squelching to actually induce a decrease in signal – not no change in signal.

4) It is extremely odd that the EC/IC50s for the ligands in the TR-FRET assay are right shifted relative to the Gal4-transactivation assays. This is inconsistent with most ligands for nuclear receptors and suggests that the ligands actually bind the receptor and affect their activity at much higher concentrations than what is reported in the cell-based assays. Importantly, there is no explanation for this. Other cell-based assays should be performed to confirm the cellular effects to determine if these potencies are indeed correct (ie gene expression).

5) It is unclear, and the authors don't expound upon why, the "inverse agonist" and "agonist" meclofenamic acid and oxaprozin, respectively, have the same responses in the TR-FRET assays? One would expect, at the very least, to have somewhat slightly differing responses between the 2 given their differing cellular responses. Granted one is less potent than the other, but one would expect their abilities to recruit cofactors to be slightly different. The same could be said about their abilities to affect NURR1 homo- and heterodimerization.

6) Given the structural differences between some of the other compounds and the fact that the authors tout this in their abstract, at least one other "inverse agonist" should have been profiled (ie valdecoxib or parecoxib) to the same extent as meclofenamic acid and oxaprozin.

Should these experiments be performed and validate these compounds as NURR1 modulators, this reviewer believes that this paper will influence how people view NURR1 and interrogate it as a therapeutic target.

There is no mention of how statistical analysis was performed, which needs to be included. Furthermore, information on how IC/EC50s were generated should also be described (i.e. was GraphPad Prism used? What version?). While there is some mention of the "n" for TR-FRET and Gal4 assays, it is unclear whether this indicates the number of wells (technical replicates) or the number of times the assay was run (biological replicates). Regardless, both need to be included.

Reviewer #3 (Remarks to the Author):

Nurr1 is an orphan (yet to have a ligand identified, should one exist) nuclear receptor that has been linked to key neurodegenerative disorders. As such, it is an interesting potential therapeutic target. Without pharmacological tools, however, it will be difficult to flesh out the true contribution of this receptor to disease. In this study, the authors attempt to address this shortcoming by exploring a series of COX inhibitors as non-specific ligands at Nurr1. Further, they investigate the impact of these ligands on co-regulator recruitment and dimerisation status of Nurr1. Overall, this paper will inform future drug discovery efforts at Nurr1 and has commendably thorough methods that will enable other researchers to also follow up these findings.

I have only a few things that I feel should be addressed by the authors:

1. the term cooperativity seems to be used imprecisely throughout the manuscript. They authors

themselves identify that the ligands, for example meclofenamic acid and amodiaquine, have additive effects at Nurr1. This is different to cooperative effects, where the affinity or potency of one ligand is altered by the other. I would recommend that more than a single concentration of ligand in Fig 3 is used so that global curve fitting can be used to determine whether cooperativity truly exists (it seems like it doesn't but this should be demonstrated).

2. The term unspecific is used in some sections of the manuscript - this should be changed to non-specific.

Dear Reviewers

We thank you very much for your time and efforts spent in reviewing our manuscript on NSAIDs as Nurr1 modulators. Your feedback was exceptionally constructive and has prompted us to perform several additional experiments to address the “unanswered questions” in our observations. We have expanded our entire screening to the two further NR4A receptors Nur77 and NOR1, and observed that all three proteins are modulated by NSAIDs which particularly on NOR1 is a valuable finding that demonstrates druggability of this nuclear receptor, as well. Moreover, we have transferred our observations from the hybrid Gal4-Nurr1 assay setting to the three human Nurr1 response elements and found that AQ and CQ act as Nurr1 agonist on all Nurr1-REs while parecoxib and oxaprozin were confirmed as inverse agonists in all settings. For MFA, we found an “intermediate” profile suggesting selective modulatory properties. With the help of further cell-free test systems, we are eventually able to clearly distinguish Nurr1 agonists and inverse agonists also on protein level. In brief, agonists promote dimerization of Nurr1 and recruitment of co-regulators while inverse agonists cause opposite effects and MFA has a Nurr1 monomer preferential profile. Overall, your constructive suggestions were very fruitful and have strongly helped expanding the impact of our study. We have addressed all your concerns (point-by-point answers are listed below) and we hope our revised manuscript meets your expectations.

Sincerely

Daniel Merk

Reviewer #1 (Remarks to the Author):

Willems et al. describe the characterization of a series of structurally diverse non-steroidal anti-inflammatory drugs as Nurr1 modulators. In the context of a Gal4-Nurr1 LBD construct, they act as agonists or inverse agonists, and biochemical studies confirm that they modulate the binding of a series of corepressor peptides to the Nurr1 LBD. These studies are interesting as far as they go but fall well short of demonstrating that the compounds are indeed agonists and inverse agonists. Three important further characterizations are needed.

We thank the reviewer for this positive and constructive feedback. We have addressed the concerns and performed several additional experiments to address the concerns.

The first is simply (and remarkably) that the effect of the compounds on the intact Nurr1 receptor has not been assessed. This could be done easily by cotransfection with a standard NBRE reporter but should also include the dimeric ER0 and IR5 response elements noted in the discussion. To more critically test their effects on the native Nurr1, the response of a range of endogenous Nurr1 target genes should also be defined, with and without Nurr1 knockdown.

Revised. As requested, we have employed NBRE (Nurr1 monomer), POMC (Nurr1 homodimer) and DR5 (Nurr1:RXR heterodimer) reporters to study the activity of Nurr1 modulators in more physiological cellular settings. These experiments have indeed enabled a markedly deeper understanding of the compounds' activity profiles. The reporter gene assays with the three human Nurr1 response elements confirmed

agonism for AQ and CQ as well as inverse agonism of parecoxib and oxaprozin while we observed a selective modulatory profile of MFA. In order to correlate these findings with Nurr1's dimerization state, we have studied the effect of all five compounds on Nurr1 homodimerization and heterodimerization by HTRF in cell-free setting. These experiments demonstrated that Nurr1 agonists promote Nurr1 dimerization while the inverse agonists and MFA decrease dimer formation. In addition, we have expanded our studies on co-regulator recruitment by Nurr1 which revealed another discriminating characteristic for agonists and inverse agonists: while AQ and CQ promote recruitment of canonical co-regulators, inverse agonists cause the opposite effect.

The second is that they have not tested whether the compounds affect the activity of the other 2 NR4A family members. This is critical for assessment of their potential biologic activity.

Revised. We have expanded the entire NSAID screening to the remaining NR4A receptors Nur77 (NR4A1) and NOR1 (NR4A3). We found that NSAIDs are active on all three NR4A receptors with very similar activity profiles providing further evidence for the close relation of these proteins. Moreover, several NSAIDs emerge as first-in-class NOR1 modulators and demonstrate that this NR4A receptor is druggable, too. Additionally, we have profiled the Nurr1 modulators used as tools in our study (MFA, parecoxib and oxaprozin) for selectivity on lipid-activated transcription factors outside the NR4A family.

Finally, this story could be significantly strengthened by using mutagenesis to probe the NSAID ligand binding site. The clearly demonstrated additive effects with amodiaquine strongly suggest that the other identified pocket mediates NSAID compound effects, but that should be tested experimentally. Ideally, the mutagenesis studies could be done in the presence and absence of amodiaquine to functionally define the distinct sites.

Revised. We have performed additional experiments to further study simultaneous Nurr1 modulation by AQ-type and NSAID-type ligands which agree with our hypothesis. Mutagenesis experiments were not included, however.

Reviewer #2 (Remarks to the Author):

Given its decreased expression in dopaminergic neurons, NURR1 appears to be an excellent target for disease modulation like Parkinson's disease. Despite initial crystal structures that did not define a clear ligand binding pocket, indicating its transcriptional activity may be ligand-independent, recent reports have suggested it may indeed be ligand-regulated. Despite this, few potent NURR1 modulators have been described. Based on a recent co-crystal structure of NURR1 bound to prostaglandin A1, Willems et al. hypothesized that COX inhibitors may be NURR1 ligands. After a screen of COX-1 and COX-2 inhibitors, they identified several structurally distinct ligands that appear to function as NURR1 agonists and inverse agonists in vitro using a Gal4 NURR1 transactivation assay. These data support the co-crystal structure data and identify COX inhibitors as potential therapeutics to exploit for further drug design.

The concepts presented in the manuscript are interesting and support previously published work indicating that PGA2 regulates NR4A2. So that fact that COX inhibitors modulate

NR4A2 is in-line with this. However, there are several points that need addressing to strengthen the conclusions:

We thank the reviewer for the critical evaluation of our manuscript, the positive response and the constructive feedback.

1) Figure 2 could be strengthened by performing knock downs of NURR1 in the cell-based assays to prove that the compounds are NURR1 selective. Furthermore, it would be ideal to screen these compounds against other nuclear receptors to determine how selective these ligands are, particularly against members of the NR4A family (NURR77 and NOR1) and other nuclear receptor subfamilies with members demonstrated to be modulated by prostaglandins – the PPARs.

Revised. As requested, we have included selectivity profiling on NR4A receptors and other lipid-activated transcription factors (PPAR, RXR, RAR). Since we discovered comparable activity on Nur77 (NR4A1) and NOR1 (NR4A3) for all Nurr1 modulators that we tested initially (AQ, CQ, MFA, parecoxib, oxaprozin), we have expanded the Nur77/NOR1 profiling to all NSAIDs in our screening collection. We found NSAIDs active on all three NR4A receptors with very similar activity profiles providing further evidence for the close relation of the NR4A proteins. Several NSAIDs emerge as first-in-class NOR1 modulators demonstrating that this protein is druggable, as well.

Regarding the requested knockdown experiments, we fully agree with the reviewer that it is very important to show that effects are Nurr1 mediated. Knockdown in the reporter gene assays make little sense to us, however. In these test systems, Gal4-Nurr1 (or full-length human Nurr1) is overexpressed by transient transfection and a knockdown would be equivalent to not transfecting the receptor expression clone at all. This would result in no (or very weak) reporter activity which in turn wouldn't allow any conclusion on the compounds' activity, especially concerning the inverse agonists which decrease the reporter signal. As an alternative way to demonstrate that the compounds have no unspecific activity in the assay setup and that their activity is Nurr1 mediated, we have performed control experiments with Gal4-VP16. This hybrid protein - as Nurr1 - is a potent transcriptional inducer and strongly stimulates reporter gene expression. We have profiled all compounds modulating reporter activity in any Gal4-NR4A assay setting in the Gal4-VP16 setting to reveal unspecific effects on reporter gene expression which (amongst others) validated MFA, parecoxib and oxaprozin as Nurr1 modulators. In addition, the HTRF based assays have demonstrated Nurr1 modulation for all these compounds in an orthogonal setting. Nurr1 mediated activity of AQ, CQ, MFA, parecoxib and oxaprozin, therefore, is beyond doubt.

2) Figure 3 is hardly indicative of the notion that the 2 ligands (meclofenamic acid and amodiaquine) are interacting with the receptor independently. Further studies need to be performed to even possibly make this claim: a) perform the transactivation assays but titrate the ligands down to significantly lower concentrations so one can see the actual intersection of the 2 ligands when they are not activating the receptor. b) perform a biochemical based study to prove the 2 binding events like isothermal calorimetry or CD thermal melt. Given that cloroquine is also a reported ligand for NURR1, similar studies should be performed to validate the effects observed with amodiaquine.

Revised. We have broadly expanded our experiments to support the hypothesis of two binding sites. As suggested, we have included further titration experiments covering also lower concentrations. In addition, we have studied the combination of AQ with the inverse agonist parecoxib which revealed non-competitive behavior. Furthermore,

we have performed the experiments with AQ/MFA and AQ/parecoxib combinations on human Nurr1 response elements, as well. The results of all these experiments support our hypothesis for two binding sites in the Nurr1 LBD.

3) The fact that the amodiaquine had “no effects” due to squelching is highly unlikely given that a) it did recruit SRC2 and SRC3 peptides (albeit weakly) at 100uM, which is consistent with previous reports for SRC3 and b) one would expect squelching to actually induce a decrease in signal – not no change in signal.

Revised. We fully agree with reviewer’s notion, of course, that it is incorrect to state that AQ has no effects, but this was not our conclusion. We also agree that from the photophysical characteristics of AQ one would expect a false decrease in the HTRF signal but increased recruitment (raise in signal) and quenching (decrease in signal) might actually add up to no change in the signal. It is undeniable that AQ (and also CQ) interfere with the HTRF system making results obtained with these compounds difficult to interpret. To overcome this issue, we have included another experimental setting where the ligand (AQ, CQ, MFA, parecoxib, oxaprozin) concentration was fixed which is less prone to cause artifact signals. In this setting we observed that AQ and CQ promote NCoR1 recruitment to Nurr1. In summary, we can conclude that agonists (AQ, CQ) promote interactions of Nurr1 with co-regulators while inverse agonists have the opposite effect and reduce Nurr1-co-regulator interactions.

4) It is extremely odd that the EC/IC50s for the ligands in the TR-FRET assay are right shifted relative to the Gal4-transactivation assays. This is inconsistent with most ligands for nuclear receptors and suggests that the ligands actually bind the receptor and affect their activity at much higher concentrations than what is reported in the cell-based assays. Importantly, there is no explanation for this. Other cell-based assays should be performed to confirm the cellular effects to determine if these potencies are indeed correct (ie gene expression).

Revised. It should be noted that the reported data from cell-free experiments are no Kd values but EC50 values. Unlike Kd, the EC50 depends on the concentration of the binding target which in the cell-free system likely is markedly higher. Moreover, the cellular and cell-free assay setups are very different test systems with different (orthogonal) readouts. In the recruitment assay, a single binding event per Nurr1 LBD is detected while the functional receptor in the cellular setting can repeatedly induce reporter gene expression. Still, the reviewer is perfectly right that further validation of the activity data is necessary. We have performed three additional cellular assay types based on the three human Nurr1 response elements which revealed potencies for the Nurr1 modulators that are consistent with their activity on Gal4-Nurr1.

5) It is unclear, and the authors don’t expound upon why, the “inverse agonist” and “agonist” meclofenamic acid and oxaprozin, respectively, have the same responses in the TR-FRET assays? One would expect, at the very least, to have somewhat slightly differing responses between the 2 given their differing cellular responses. Granted one is less potent than the other, but one would expect their abilities to recruit cofactors to be slightly different. The same could be said about their abilities to affect NURR1 homo- and heterodimerization.

Revised. We thank the reviewer for raising this point. It was indeed unclear how agonist and inverse agonists differentially regulate Nurr1 activity regarding protein-protein interactions. We have broadly expanded our experiments in this direction and now can clearly distinguish between agonist and inverse agonist effects. In brief, agonists promote homo- and heterodimerization of Nurr1 as well as Nurr1 interactions with co-regulators. Inverse agonists have opposite effects and diminish dimerization

as well as co-regulator recruitment. We feel that these experiments and the conclusions drawn from their results significantly improve the outcome of our study and markedly enhance our understanding of Nurr1 modulation by ligands.

6) Given the structural differences between some of the other compounds and the fact that the authors tout this in their abstract, at least one other “inverse agonist” should have been profiled (ie valdecoxib or parecoxib) to the same extent as meclofenamic acid and oxaprozin.

Revised. We agree that the impact of our observations is strengthened by expanding them to another inverse agonist chemotype. As suggested, we have included parecoxib in all experiments.

Should these experiments be performed and validate these compounds as NURR1 modulators, this reviewer believes that this paper will influence how people view NURR1 and interrogate it as a therapeutic target.

We thank the reviewer for this feedback. Your suggestions were very constructive and we have addressed them in full with additional experiments.

There is no mention of how statistical analysis was performed, which needs to be included. Furthermore, information on how IC/EC50s were generated should also be described (i.e. was GraphPad Prism used? What version?). While there is some mention of the “n” for TR-FRET and Gal4 assays, it is unclear whether this indicates the number of wells (technical replicates) or the number of times the assay was run (biological replicates). Regardless, both need to be included.

Revised. We thank the reviewer for raising this important point. We have added a paragraph on statistical evaluation to the methods section clarifying all these points.

Reviewer #3 (Remarks to the Author):

Nurr1 is an orphan (yet to have a ligand identified, should one exist) nuclear receptor that has been linked to key neurodegenerative disorders. As such, it is an interesting potential therapeutic target. Without pharmacological tools, however, it will be difficult to flesh out the true contribution of this receptor to disease. In this study, the authors attempt to address this shortcoming by exploring a series of COX inhibitors as non-specific ligands at Nurr1. Further, they investigate the impact of these ligands on co-regulator recruitment and dimerisation status of Nurr1. Overall, this paper will inform future drug discovery efforts at Nurr1 and has commendably thorough methods that will enable other researchers to also follow up these findings.

We thank the reviewer very much for this positive response on our manuscript.

I have only a few things that I feel should be addressed by the authors:

1. the term cooperativity seems to be used imprecisely throughout the manuscript. They authors themselves identify that the ligands, for example meclofenamic acid and amodiaquine, have additive effects at Nurr1. This is different to cooperative effects, where the affinity or potency of one ligand is altered by the other. I would recommend that more than a single concentration of ligand in Fig 3 is used so that global curve fitting can be used to determine whether cooperativity truly exists (it seems like it doesn't but this should be demonstrated).

Revised. We thank the reviewer for raising this point. We have changed the wording throughout the manuscript to avoid misinterpretation. Cooperativity cannot be stated from our observations, indeed. Instead, we argue that NSAID-type and AQ-type ligands modulate Nurr1 through different sites in the Nurr1 LBD simultaneously.

2. The term *unspecific* is used in some sections of the manuscript - this should be changed to *non-specific*.

Revised. The term “*unspecific*” has been replaced by “*non-specific*”.

Reviewer #1 (Remarks to the Author):

The substantial revisions effectively address prior concerns.

Reviewer #2 (Remarks to the Author):

In this revised report, Willems et al., identified several structurally distinct ligands that appear to function not only as NURR1 ligands, but as NR4A family member ligands, functioning as both agonists and inverse agonists in vitro. Based on reviewers' suggestions, they expanded their initial identification from NURR1 to include all members of the NR4A subfamily to determine that ligands modulate (basically) all receptors. They also greatly expanded their assay repertoire from Gal4 NURR1 transactivation assays to full-length transactivation assays, to demonstrate that these ligands affect NURR1 activity in a variety of conformations (homodimers, heterodimers). These data identify COX inhibitors as potential therapeutics to exploit for further drug design.

While the initial scope of the paper was to identify NURR1-selective ligands, and the data ended up describing NR4A modulating ligands, that is OK given the biology the group has uncovered surrounding NURR1 as well as the large number of scaffolds identified which can be used as starting materials for future structure activity relationship studies to develop more selective NURR1 modulators. Therefore, the paper should influence how people view NURR1 and interrogate it as a therapeutic target.

While almost all of the critiques were addressed, there are a few outstanding issues that should be addressed. These, I believe, can be addressed through modifications of the text and/or a figure. These include the following:

Figure 3. The statement "The activators AQ and CQ robustly induced reporter activity on all response elements confirming their Nurr1 agonism" seems slightly misleading based on the graphs presented in Figure 3b-d. While it is obvious that AQ robustly activated these response elements, and Table 1 indicates that CQ does give some response, fold change vs EC50 don't always correlate. Based on the way the graphs are presented, it looks like CQ has no activity at all. If CQ does robustly activate these response elements, please present the data in a way that reflects it. Otherwise, please change the wording of this sentence.

For Figure 3g & h: Again, please be careful with the wording. Authors state that the NURR1 agonists promoted formation of homo- and heterodimers. However, looking at the graphs presented, CQ mainly formed homodimers whereas AQ was able to form both homo and heterodimers. This suggests that the ligands have different effects on the dimerization of NURR1 which would account for some of their different cellular effects in vitro.

Figure 4. Be careful with wording of this figure as well. This also ties into the data presented in Figure 3. For example, authors state that "The co-regulator recruitment profile of inverse Nurr1 agonists parecoxib and oxaprozin slightly differed from that of MFA in terms of potencies but both drugs exhibited displacement of NCoR-1, SMRT, PRIIPRAP and RIP140 from Nurr1" when in fact Paracoxib only displaced 2 cofactors (NCOR and SMRT). The sentence is slightly misleading.

Authors also state "the Nurr1 agonists AQ and CQ revealed a tendency to promote recruitment of NCoR-1, SMRT, PRIIPRAP and RIP140". This statement seems to indicate that both ligands lead to recruitment of all 4 peptides to NURR1. Based on graphs in 4b-e, AQ did not recruit any cofactors whereas CQ only recruited all but SMRT 2. In fact, the only data supporting that AQ can recruit any cofactor is figure 4f – where NCOR1 is demonstrated to be recruited to NURR1 via AQ. Please fix the wording of both of these statements to reflect the data presented both in the results section and the discussion. If anything, these data could potentially explain why AQ has the better agonistic effect on NURR1 than CQ in transactivation assays.

Minor point: What is D22? (It is clearly a peptide, but it does appear to be slightly modulated by ligand treatment, yet there is no mention of it?)

Finally, the authors did an excellent job of addressing previous concerns regarding how statistical analysis was performed, including information on how IC/EC50s were generated, including the number of replicates for assays, etc.

Dear Reviewers

We thank you very much for the careful evaluation of our manuscript and the constructive criticism. Your comments have clearly helped to improve the quality of our study. Point-by-point answers on your concerns are listed below.

Sincerely

Daniel Merk

REVIEWERS' COMMENTS:

Reviewer #1 (Remarks to the Author):

The substantial revisions effectively address prior concerns.

We thank the reviewer very much for reevaluating our manuscript and for the positive response.

Reviewer #2 (Remarks to the Author):

In this revised report, Willems et al., identified several structurally distinct ligands that appear to function not only as NURR1 ligands, but as NR4A family member ligands, functioning as both agonists and inverse agonists in vitro. Based on reviewers' suggestions, they expanded their initial identification from NURR1 to include all members of the NR4A subfamily to determine that ligands modulate (basically) all receptors. They also greatly expanded their assay repertoire from Gal4 NURR1 transactivation assays to full-length transactivation assays, to demonstrate that these ligands affect NURR1 activity in a variety of conformations (homodimers, heterodimers). These data identify COX inhibitors as potential therapeutics to exploit for further drug design. While the initial scope of the paper was to identify NURR1-selective ligands, and the data ended up describing NR4A modulating ligands, that is OK given the biology the group has uncovered surrounding NURR1 as well as the large number of scaffolds identified which can be used as starting materials for future structure activity relationship studies to develop more selective NURR1 modulators. Therefore, the paper should influence how people view NURR1 and interrogate it as a therapeutic target. While almost all of the critiques were addressed, there are a few outstanding issues that should be addressed. These, I believe, can be addressed through modifications of the text and/or a figure. These include the following:

We thank the reviewer very much for evaluating our manuscript a second time and for the positive and constructive response. We agree with all further concerns that have been raised. Point-by-point answers are listed below.

Figure 3. The statement "The activators AQ and CQ robustly induced reporter activity on all response elements confirming their Nurr1 agonism" seems slightly misleading based on the graphs presented in Figure 3b-d. While it is obvious that AQ robustly activated these response elements, and Table 1 indicates that CQ does give some response, fold change vs

EC50 don't always correlate. Based on the way the graphs are presented, it looks like CQ has no activity at all. If CQ does robustly activate these response elements, please present the data in a way that reflects it. Otherwise, please change the wording of this sentence.

Revised. We thank the reviewer for raising this important point. CQ does activate Nurr1 but its activation efficacy is markedly lower than for AQ in the full-length Nurr1 reporter settings. We have updated the respective section in the text to point this out and we have added Supplementary Figure 3 showing the dose-response curves for CQ at an appropriate scaling.

For Figure 3g & h: Again, please be careful with the wording. Authors state that the NURR1 agonists promoted formation of homo- and heterodimers. However, looking at the graphs presented, CQ mainly formed homodimers whereas AQ was able to form both homo and heterodimers. This suggests that the ligands have different effects on the dimerization of NURR1 which would account for some of their different cellular effects in vitro.

Revised. Again, we thank the reviewer for pointing this out. There is indeed a difference in the abilities of AQ and CQ to modulate dimerization. However, it's the other way around: AQ preferably induces homodimerization while CQ induced both homo- and heterodimer formation. We have updated the respective section and included this aspect in the discussion.

Figure 4. Be careful with wording of this figure as well. This also ties into the data presented in Figure 3. For example, authors state that "The co-regulator recruitment profile of inverse Nurr1 agonists parecoxib and oxaprozin slightly differed from that of MFA in terms of potencies but both drugs exhibited displacement of NCoR-1, SMRT, PRIPRAP and RIP140 from Nurr1" when in fact Paracoxib only displaced 2 cofactors (NCOR and SMRT). The sentence is slightly misleading.

Revised. We have revised the wording of this section, too, to describe the observed results more clearly.

Authors also state "the Nurr1 agonists AQ and CQ revealed a tendency to promote recruitment of NCoR-1, SMRT, PRIPRAP and RIP140". This statement seems to indicate that both ligands lead to recruitment of all 4 peptides to NURR1. Based on graphs in 4b-e, AQ did not recruit any cofactors whereas CQ only recruited all but SMRT 2. In fact, the only data supporting that AQ can recruit any cofactor is figure 4f – where NCOR1 is demonstrated to be recruited to NURR1 via AQ. Please fix the wording of both of these statements to reflect the data presented both in the results section and the discussion. If anything, these data could potentially explain why AQ has the better agonistic effect on NURR1 than CQ in transactivation assays.

Revised. We agree with the reviewer that the titration of AQ in the recruitment assays does not point to recruitment of any co-regulator (in contrast to CQ which indeed induced recruitment of all 4 peptides although the effect on SMRT is slightly weaker). However, the photophysical characteristics of AQ must be considered here and while it cannot be stated that AQ affects interactions of Nurr1 with co-regulators, the opposite statement can neither be made. The apparent affinity of Nurr1 to NCoR-1 is

increased by a fixed concentration of AQ which rather points to enhanced recruitment. We have updated the respective section and discussion on this matter accordingly.

Minor point: What is D22? (It is clearly a peptide, but it does appear to be slightly modulated by ligand treatment, yet there is no mention of it?)

Revised. There is indeed an effect of Nurr1 ligands on Nurr1-D22 interactions. However, D22 is an artificial peptide and, therefore, was not further considered in the dose-response characterization. We have added a sentence to clarify this.

Finally, the authors did an excellent job of addressing previous concerns regarding how statistical analysis was performed, including information on how IC/EC50s were generated, including the number of replicates for assays, etc.

We thank the reviewer again for the very careful evaluation of our manuscript, the constructive criticism and the positive response.